. Pathogens

# Immune signatures of SARS-CoV-2 infection resolution in human lung tissues

Devin Kenney[1,2], Aoife K. O'Connell[1,2,3], Anna E. Tseng[1,2,3], Jacquelyn Turcinovic[1,2,4], Maegan L. Sheehan[5☯], Adam D. Nitido[5☯], Paige Montanaro[2,3], Hans P. Gertje[2,3], Maria Ericsson[6], John H. Connor[1,2], Vladimir Vrbanac[5], Nicholas A. Crossland[1,2,3], Christelle Harly[7☯*], Alejandro B. Balazs[5☯*], Florian Douam [1,2☯*]

1 Department of Virology, Immunology, and Microbiology, Boston University Chobanian & Avedisian School of Medicine, Boston, Massachusetts, United States of America, 2 National Emerging Infectious Diseases Laboratories, Boston University, Boston, Massachusetts, United States of America, 3 Department of Pathology and Laboratory Medicine, Boston University Chobanian & Avedisian School of Medicine, Boston, Massachusetts, United States of America, 4 Bioinformatics Program, Boston University, Boston, Massachusetts, United States of America, 5 Ragon Institute of MGH, MIT and Harvard, Cambridge, Massachusetts, United States of America, 6 Electron Microscopy Core Facility, Harvard Medical School, Boston, Massachusetts, United States of America, 7 Nantes Université, Inserm UMR 1307, CNRS UMR 6075, Université d'Angers, CRCI2NA, Nantes, France

☯ These authors contributed equally to the work.
* fdouam@bu.edu (FD); abalazs@mgh.harvard.edu (ABB); christelle.harly@univ-nantes.fr (CH)

## Abstract

While human autopsy samples have provided insights into pulmonary immune mechanisms associated with severe viral respiratory diseases, the mechanisms that contribute to a clinically favorable resolution of viral respiratory infections remain unclear due to the lack of proper experimental systems. Using mice co-engrafted with a genetically matched human immune system and fetal lung xenograft (fLX), we mapped the immunological events defining successful resolution of SARS-CoV-2 infection in human lung tissues. Viral infection is rapidly cleared from fLX following a peak of viral replication, histopathological manifestations of lung disease and loss of AT2 program, as reported in human COVID-19 patients. Infection resolution is associated with the activation of a limited number of hematopoietic subsets, including inflammatory monocytes and CD3-expressing macrophage-like cells, which are highly enriched in viral RNA and dissipate upon infection resolution. Specific human fibroblast and endothelial subsets also elicit robust antiviral and monocyte chemotaxis signatures, respectively. Notably, systemic depletion of human CD4 + cells, but not CD3 + cells, significantly abrogates infection resolution in fLX and induces persistent infection, supporting the dominant role of peripheral CD4 + monocytes over T-cells in the resolution of acute SARS-CoV-2 infection. Collectively, our findings unravel a comprehensive picture of the immunological events defining effective resolution of SARS-CoV-2 infection in human lung tissues, revealing markedly divergent immunological trajectories between resolving and fatal COVID-19 cases.

**Data availability statement:** The Genome Expression Omnibus (GEO) accession number to access the raw data of our scRNA-seq analysis is GSE255200.

**Funding:** This work was supported in part by funds or awards from Boston University (www.bu.edu; NEIDL start-up funds and Peter Paul Career Development Professorship to F.D.), the National Institutes of Health (www.nih.gov; K22 AI144050 to F.D.; UL1 TR001430 to F.D. and N.A.C.; R01AI174875, R01AI174276, DP1DA060607 and DP2DA040254 to A.B.B.; S10 OD026983 and S10OD030269 to N.A.C.), the U.S. Center of Disease Control (www.cdc.gov; subcontract 200-2016-91773-T.O.2 to A.B.B.), the Evergrande Massachusetts Consortium on Pathogenesis Readiness (MassCPR) (https://masscpr.hms.harvard.edu; to NEIDL as a university center, and to A.B.B.), the INSERM, CNRS and Université de Nantes (www.inserm.fr; www.cnrs.fr; www.univ-nantes.fr to C.H.), the Fondation pour la Recherche Médicale (www.frm.org; DEQ20170839118 to C.H.), and the French National Research Agency (www.anr.fr; LabEX IGO, ANR-11-LABX-0016 to C.H.). D.K. and A.E.T. were supported by a NIH (www.nih.gov) T32 training grant in immunology (T32AI007309). Salaries were supported by Boston University (F.D.), the National Institute of Health (D.K., A.E.T., A.B.B., F.D.) and CNRS (C.H.). The funders had no role in study design, data collection and analysis, decision to publish, or preparation of the manuscript.

**Competing interests:** I have read the journal's policy and the authors of this manuscript have the following competing interests: A.B.B. is a founder of Cure Systems LLC. The other authors declare no competing interests in relation to this study.

## Author summary

COVID-19 can result in a wide range of clinical outcomes, from asymptomatic infection to death. While the cellular and molecular correlates associated with fatal COVID-19 have been well characterized, the immunological underpinnings of effective resolution of acute infection in humans have remained elusive due to the lack of experimental systems recapitulating human immune responses to SARS-CoV-2, the etiologic agent of COVID-19. In this study, we used mice co-engrafted with a human immune system and human lung tissues to unveil human immune correlates of acute SARS-CoV-2 infection resolution in human lung tissues. We discovered that this process is associated with the emergence and antiviral responses of two specific immune cell populations: a particular inflammatory monocyte subpopulation and CD3-expressing macrophage-like cells, both of which are highly enriched in viral RNA. Consistent with these findings, systemic depletion of circulating monocytes abrogated infection resolution and resulted in signatures of chronic infection both systemically and in human lung tissues. Collectively, our study provides an original picture of the various immunological actors involved in the resolution of SARS-CoV-2 infection in human lung tissues, opening avenues to a deeper understanding of the factors governing differential susceptibility to SARS-CoV-2 infection across individuals.

## Introduction

Coronavirus disease 2019 (COVID-19) is a respiratory disease that has swept the world since its emergence in the Wuhan province of China in late 2019. The etiologic agent of COVID-19, the Severe Acute Respiratory Syndrome Coronavirus 2 (SARS-CoV-2) is a plus-sense, enveloped RNA virus that targets the epithelium of the respiratory tract. Infection results in varying severities of COVID-19, with most cases being mild to asymptomatic. The onset of severe disease is associated with aberrant immune responses (e.g., excessive pulmonary infiltration of myeloid cells, inflammasome-activated monocytic cells, macrophage exacerbated inflammation) and severe lung injury (e.g., lung consolidation, diffuse alveolar damage (DAD), and thrombosis) [1–9].

A large number of human studies have been instrumental in unraveling cellular and molecular processes driving severe COVID-19 disease in infected tissues, particularly in the respiratory tract, using autopsy samples [10–13]. In parallel, human studies of resolving COVID-19, including controlled human challenge studies, have leveraged peripheral blood and nasopharyngeal samples to identify human signatures of effective infection resolution and mild disease [14–16]. Notably, this includes individuals with specific HLA haplotypes [15], evidence of previous coronavirus exposure [15,17,18], rapid nasopharyngeal immune infiltration [14], and non-productive infection of nasopharyngeal T-cells and macrophages [14]. However, tissue-specific human immunological processes associated with protection, such as extravasation

of recruited immune lineages and their differentiation processes, have remained elusive due to the ethical considerations associated with tissue sampling of individuals with mild disease and human challenge models. Although large and small animal models of COVID-19 are available, the high-cost and limited reagent availability associated with non-human primate models, and the large divergence between rodent and human immune systems [19,20] further underscore the need for additional models capable of recapitulating human protective immune responses to SARS-CoV-2 infection.

Mice engrafted with human fetal lung xenograft (fLX) support infection by multiple human respiratory viruses, including human cytomegalovirus (HCMV), Middle East respiratory syndrome coronavirus (MERS-CoV) and SARS-CoV-2 [21–23]. Upon co-engraftment with a human immune system (HIS), these animals also mount lung-resident human immune responses against these pathogens [21,23–25]. Recently, our group reported that mice engrafted with fLX are highly susceptible to SARS-CoV-2 infection and lung tissue damage and support persistent viral infection [23]. However, co-engraftment of fLX and HIS in a xenorecipient strain supporting enhanced myelopoiesis (i.e., HNFL mouse model) rapidly blunted viral infection and prevented widespread acute viral replication across fLX, resulting in protection from histopathology of fLX [23]. Our findings unraveled a human macrophage antiviral signature as a correlate of such rapid protection against SARS-CoV-2 infection. However, how the human immune system mobilizes and resolves infection upon extensive viral replication within human lung tissues, a context that more likely describes mild cases of SARS-CoV-2 infection, has remained elusive.

In this study, we leverage previously described immunodeficient mice engrafted with a human fetal lung xenograft (fLX) and a genetically matched human immune system, fetal liver and thymus (BLT-L mice) [21] to conduct the first comprehensive mapping of human immunological correlates of resolution of acute SARS-CoV-2 infection in human lung tissues. Consistent with previous reports [24,25], BLT-L mice are permissive to SARS-CoV-2 infection following direct viral inoculation into the fLX. Infection swiftly resolves by 6 days post-infection (dpi) following an early viral replication peak at 2 dpi and is defined by the emergence of a limited set of hyper-activated hematopoietic subsets, including inflammatory monocytes and a unique cell population that expresses *CD3* and genes canonically associated with macrophages. Of these, inflammatory monocytes appear to mount the most robust antiviral responses and are highly enriched in viral RNA before dissipating from fLX after infection resolution. Specific fibroblast and endothelial cell subsets also exhibit antiviral and hematopoietic chemotaxis signatures, respectively, during acute infection. At 12 dpi, the immune landscape in fLX is characterized by an increase in CD4 + patrolling monocytes, conventional dendritic cells and CD206 + interstitial macrophages (IM). Notably, systemic depletion of human CD4 + cells, but not human CD3 + cells, significantly abrogates SARS-CoV-2 clearance in fLX and causes persistent infection, highlighting the prevailing role of CD4 + circulating monocytes over T-cells in resolving acute SARS-CoV-2 infection.

Collectively, our work sheds light on a unique set of immunological events associated with SARS-CoV-2 resolution in human lung tissues. This work opens avenues to enhance our understanding of the hematopoietic and non-hematopoietic mechanisms of lung infection resolution, which could inform the development of immunotherapies against viral respiratory diseases.

## Results

### BLT-L mice effectively clear SARS-CoV-2 infection following acute viral replication

Previous work from our laboratory [23] and others [21,24,25] have shown that immunodeficient mice can successfully be engrafted with human fetal lung xenograft (fLX) alone or in combination with a human immune system (HIS). In this study, we leveraged a previously reported mouse model co-engrafted with fetal liver and thymus as well as with human hematopoietic stem cells (HSC) and fLX (BLT-L mice) [21]. Fetal liver and thymus were engrafted under the renal capsule of adult NOD.Cg-*Prkdc^scid^ Il2rg^tm1Wjl^*/SzJ (NSG) mice (12–16 weeks old) prior to intravenous HSC injection. A piece of fetal lung tissue was subcutaneously engrafted on the flank of each animal, as described previously [21] (**Fig 1A**). To determine the susceptibility of BLT-L mice to SARS-CoV-2 infection and their ability to effectively clear infection, BLT-L mice were

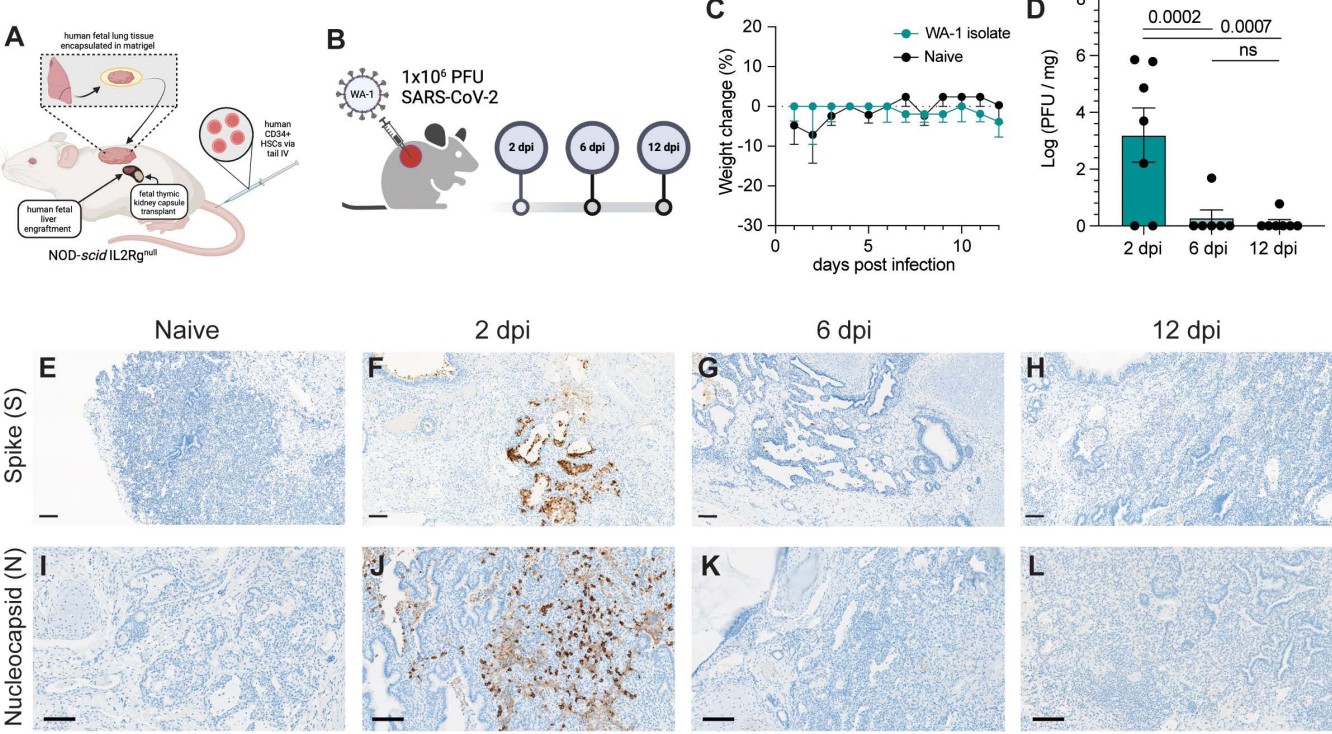

**Fig 1. BLT-L mice effectively resolve SARS-CoV-2 infection. (A)** Representative schematic of BLT-L mice. Created with BioRender.com. **(B)** Fetal lung xenografts (fLX) of BLT-L mice were infected with 1x10⁶ PFU of SARS-CoV-2 WA-1 isolate via subcutaneous intra-graft injection. Created with BioRender.com. **(C)** BLT-L mice were monitored for weight change over the course of 12 days post-infection (dpi). **(D)** Viral titer (log(PFU/mg)) in fLX at 2, 6, and 12 dpi as determined by plaque assay. **(E-H)** Immunohistochemistry for SARS-CoV-2 S and **(I-L)** N protein was performed on naïve fLX **(E,I)** and fLX at 2 dpi **(F,J)**, 6dpi **(G,K)**, and 12 dpi **(H,L)**. Scale bar = 100 μM. Data are representative of two to three independent cohorts and donors. n = 6-10 per timepoint. Error bars represent mean ± standard error of the mean. One-way ANOVA analysis was performed. p-values are indicated. n.s. = non-significant.

inoculated with SARS-CoV-2 (2019-nCoV/USA_WA1/2020) via intra-graft injection. We used a viral dose (10⁶ PFU) that we previously established to drive robust and persistent infection in fLX of immunodeficient mice not engrafted with a HIS [23] (Fig 1B).

Throughout the course of infection, mice did not display any weight loss (Fig 1C) or clinical signs of disease such as lethargy or lack of responsiveness. To assess lung histopathology and viral titers longitudinally, fLX were collected at 2, 6, and 12 dpi. Plaque assay was performed on fLX homogenates to determine viral titers. A significant amount of infectious viral particles could be recovered from fLX at 2 dpi (3.20 ± 2.52 log([PFU/mg of tissue]), but not at 6dpi (0.281 ± 0.688 log[PFU/mg of tissue]) and 12 dpi (0.111 ± 0.293 log[PFU/mg of tissue]) (Fig 1D). These data demonstrated a peak of viral infection at 2 dpi, prior to resolution of infection by 6 dpi. Immunohistochemistry (IHC) for SARS-CoV-2 spike (S) protein revealed infection was mainly found in the alveolar epithelium of fLX at 2 dpi (Fig 1E,1F). Consistent with viral quantification, most viral antigen was cleared by 6 dpi and became undetectable by 12 dpi (Fig 1G,1H). SARS-CoV-2 S was primarily detected in the alveolar and bronchiole epithelium along with necrotic cellular debris, consistent with the primary cell targets of SARS-CoV-2 (Fig 1F). These findings were confirmed through IHC for SARS-CoV-2 nucleoprotein (N) at 2, 6 and 12 dpi (Fig 1I-1L) and transmission electron microscopy (TEM) imaging of fLX at 2 dpi (S1A–S1C Fig). Notably, TEM substantiated evidence of productive fLX infection, as indicated by the presence of viral particles in the cytosol of epithelial cells and budding/invagination of virions (S1A–S1C Fig).

Next, we examined histopathological phenotypes associated with active and resolved infection. Interpretation of hematoxylin and eosin (H&E) staining illustrated denuding of pneumocytes, neutrophil infiltration, edema, hemorrhage, thrombosis, and pneumocyte necrosis, which correlated with sites of infection at 2 dpi (Fig 2A-2E). No major signs of histopathological lung damage were observed at 6 and 12 dpi compared to naïve fLX, indicating that fLX can mount repair mechanisms upon resolution of infection (Fig 2F-2G). A previously described semi-quantitative histopathological scoring system [23] provided statistical confirmation for a significant increase in lung pathology at 2 dpi, which was no longer apparent at 6 (Fig 1H, S1 Table) or 12 dpi. Of note, minor lung pathology was observed at baseline (naïve), likely reflecting limited graft vs. host disease. Together, infection of fLX of BLT-L mice recapitulates many important hallmarks of acute SARS-CoV-2 infection, including viral replication and histopathological manifestations of disease, prior to effective viral clearance and lung tissue repair.

## Humoral responses do not drive SARS-CoV-2 clearance in BLT-L mice despite evidence of spike selective pressure

We first asked whether SARS-CoV-2 infection resolution was driven by human neutralizing humoral responses. Consistent with the rapid clearance of infectious viral particles by 6 dpi and the known caveat that humanized mice mount limited humoral responses [26], there were no detectable neutralizing antibodies in serum collected at 2 or 12 dpi (S2A, S2B Fig). However, interestingly, genomic sequencing of virus isolated from fLX at 2 dpi revealed the selection of two

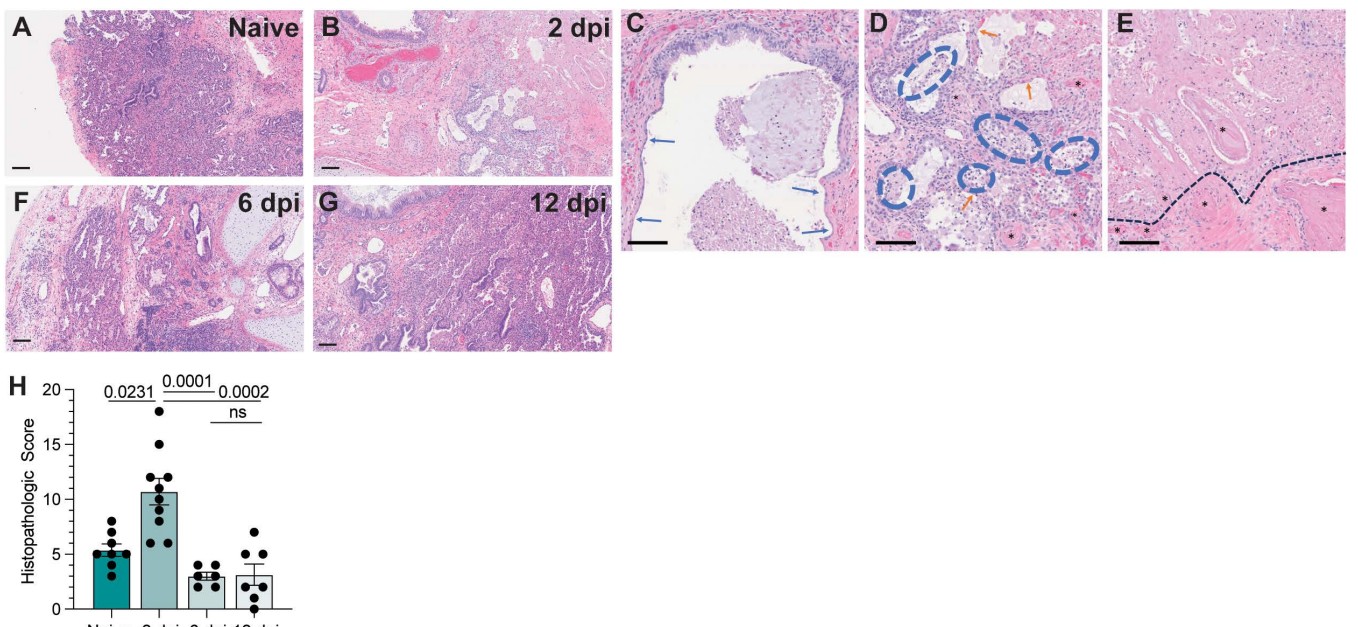

**Fig 2. BLT-L mice resolve histopathological damage in fLX. (A-E)** Hematoxylin and eosin staining was performed on naïve fLX **(A)** and fLX at 2 dpi **(B-E)**, 6dpi **(F)**, and 12 dpi **(G)**. Scale bar = 100 μM. **C:** Bronchiolar attenuation and denuding (blue arrows) correlate directly with the presence of SARS-CoV-2 spike protein. **D:** Alveolar spaces are multifocally filled with necrotic debris admixed with neutrophils and edema (blue hashes). Type II pneumocytes (ATII) are often denuded or attenuated in areas of inflammation that correlate directly with the presence of SARS-CoV-2 spike protein. Fibrin thrombi are routinely observed in neighboring parenchyma (black asterisks). **E:** Coagulative necrosis as evidenced by loss of cellular detail and generalized eosinophilia (above the black hashed line) with numerous regional fibrin thrombi (black asterisks). Although SARS-CoV-2 spike protein is not observed in the area of coagulative necrosis, viral antigens are located within adjacent tissues. **(H)** Histopathological scoring of naïve fLX and fLX at 2, 6, and 12 dpi. Data are representative of two to three independent cohorts and donors. n = 6 -10 per timepoint. Error bars represent mean ± standard error of the mean. *One-way ANOVA analysis was performed. p-values are indicated. n.s. = non-significant.*

stable mutations in 75% of fLX (S2C Fig, S2 Table). Both mutations were located in the spike N-terminal domain (NTD): an insertion (216KLRS) and a non-synonymous mutation (R245H), neither of which were present in the inoculum (S2C Fig, S2 Table). Interestingly, these two mutations were found together in 100% of the viral sequences, suggesting potential co-evolution (S2D Fig). They have also been reported as positively selected in the context of sub-optimal neutralizing antibody concentration (216KLRS) [27] or the context of cross-species adaptation (216KLRS and R245H) [28]. Despite lacking humoral responses, these findings reveal that BLT-L mice can recapitulate host-pathogen interactions that drive the positive selection of known spike adaptive mutations.

## SARS-CoV-2 infection remodels the human lung cellular environment

To comprehensively map the fLX responses associated with resolution of SARS-CoV-2 infection, we performed single-cell RNA sequencing (scRNA-seq) on fLX from naïve BLT-L mice and at 2 and 12 dpi. While most cells detected in fLX by scRNA-seq were human, a minor population of mouse cells was detected, which were excluded from downstream analysis (S3A–S3C Fig). Initial analysis of the human compartment revealed diverse hematopoietic (T-cells/innate lymphoid cells (ILC), B cells, myeloid cells and mast cells) and non-hematopoietic lineages (ciliated and non-ciliated epithelial, endothelial, mesenchymal cells and chondrocytes) in both naïve and infected fLX (Figs 3A–3C and S3D, S3E).

T-cells and innate lymphoid cells (ILC) have the same transcriptional programs, express TCR genes, and play similar effector functions [29]. Their shared features render them very challenging to distinguish by scRNA-seq in small datasets and without TCR sequencing information [26]. We thus considered them together as part of a T-cell/ILC population. T-cell/ILC frequency dramatically decreased in fLX upon infection (naïve, 66.8%; 2 dpi, 35.3%; 12 dpi, 21.8%) (Fig 3B,3C). Notably, lymphopenia was not observed in the peripheral blood of BLT-L mice, suggesting that the decrease in lymphocyte

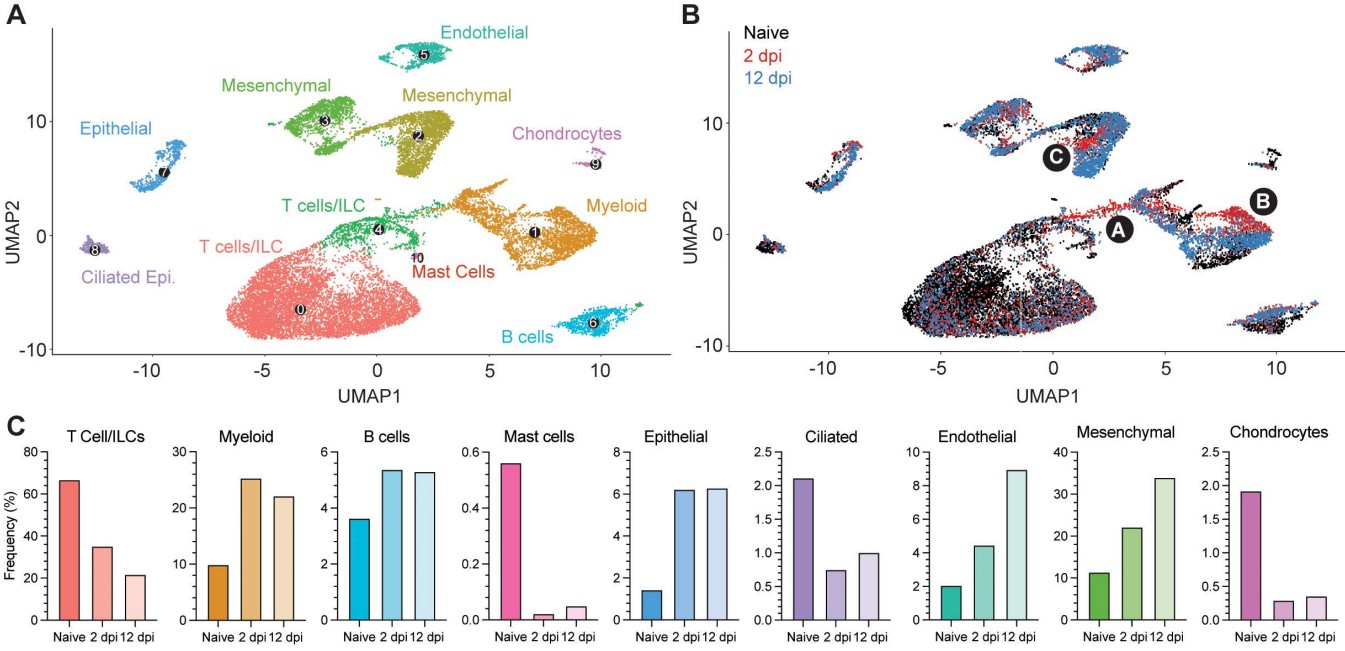

**Fig 3. Cellular remodeling in fLX upon SARS-CoV-2 infection.** Single-cell RNA sequencing analysis was performed on single-cell suspensions of naïve fLX and fLX at 2 and 12 dpi. Naïve n = 3 fLX (9,605 cells), 2 dpi n = 2 fLX (4,405 cells), and 12 dpi n = 2 fLX (5,857 cells). **(A)** UMAP plot clustering of the human cell compartment of naïve fLX and fLX at 2 and 12 dpi. **(B)** Temporal annotation of the human clusters on the UMAP plot: naïve (black), 2 dpi (red), 12 dpi (blue). Subclusters unique to 2 dpi are annotated by A (T cell compartment), B (myeloid compartment) and C (mesenchymal compartment). **(C)** Frequency of each cell compartment determined by single-cell RNA-sequencing.

PLOS Pathogens

frequencies in fLX was not directly attributed to declining circulating peripheral lymphocytes (S3F Fig). In contrast, myeloid (naïve, 9.94%; 2 dpi, 25.3%; 12 dpi, 22.2%) and B-cell subsets (naïve, 3.64%; 2 dpi, 5.38%; 12 dpi, 5.31%) relatively expanded upon infection (Fig 3B,3C). The epithelial, endothelial and mesenchymal cell frequencies increased upon infection, while mast cell, ciliated cell and chondrocyte frequencies decreased (Fig 3B,3C). Notably, most human clusters showed temporal segregation between naïve fLX and fLX at 2 and 12 dpi (Figs 3B and S3E; naïve: black subclusters; 2 dpi: red subclusters and 12 dpi: blue subclusters). This suggests that infection alters the transcriptional state of many cell types and/or drives the emergence of novel cell subsets. Notably, acute infection at 2 dpi led to the emergence of distinct T cell, myeloid, and mesenchymal cell populations, labeled as subclusters A, B, and C, respectively (Fig 3B). Resolved infection (12 dpi) was associated with the emergence of transcriptomically distinct epithelial, endothelial, mesenchymal, B-cell and myeloid subclusters; while we did not observe the emergence of transcriptomically divergent subclusters within the T-cell lineages at that time point. Temporal separation of several human lineages suggests a two-step tissue remodeling process during infection involving i) an initial antiviral phase mediated by a limited set of human subpopulations and ii) a tissue repair phase involving a broader range of human subpopulations.

Next, we wished to utilize scRNA-seq to determine the human cellular compartments enriched in viral RNA. We found that viral RNA was mainly within three major lineages: mesenchymal, T-cell/ILCs and myeloid (Fig 4A, 4B). AT2 and the overall epithelial compartment were not significantly enriched in viral RNA despite histological evidence of epithelial infection in fLX at 2 dpi (Figs 4A, 1F and 1J). Previous evidence that SARS-CoV-2 induces significant cytopathic damage

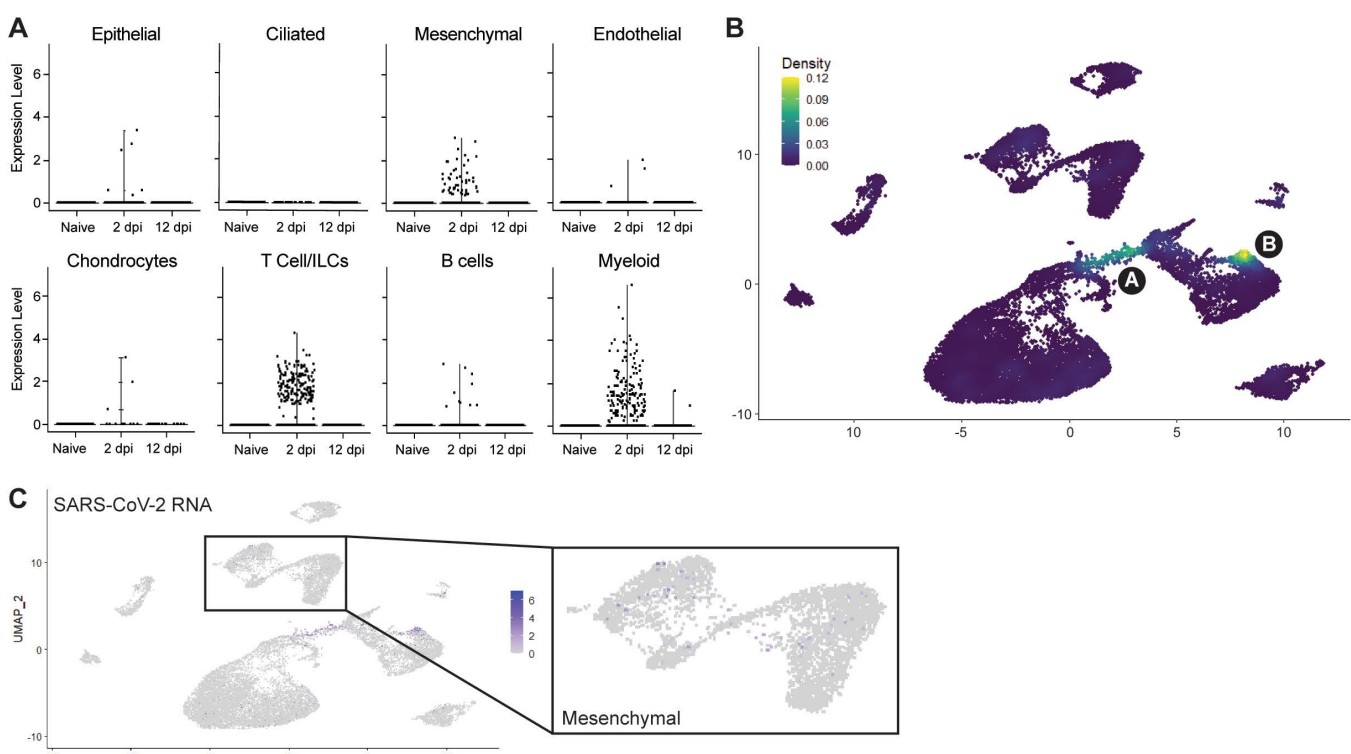

**Fig 4. Cellular compartmentalization of viral RNA during infection. (A)** Dot plots displaying the expression of SARS-CoV-2 viral RNA transcripts across the different human clusters and time points analyzed by scRNA-seq. **(B)** UMAP plot showing the distribution of SARS-CoV-2 viral RNA transcripts by density across all human cell clusters and time points analyzed by scRNA-seq. **(C)** UMAP plot showing the distribution of SARS-CoV-2 viral RNA reads across all human cell clusters and time points analyzed by scRNA-seq. Inset showing distribution of viral RNA reads in the mesenchymal cluster.

in AT2 cells [30] may suggest that infected AT2 cells are lost when undergoing the scRNA-seq pipeline. Within the mesenchymal cluster, viral reads were limited and distributed sporadically without significant enrichment within a given subcluster (Fig 4B,4C). However, viral RNA was strongly enriched within the 2 dpi-specific populations in the myeloid and T-cell clusters, previously labeled as clusters A and B (Fig 4A, 4B), warranting further investigations.

As we detected three mouse clusters, we also investigated the identity of these clusters and whether they may also associate with viral RNA. These clusters were identified as monocyte/macrophages (cluster 0, S4A, S4B Fig), mesenchymal cells with a chondrocyte-like fibroblasts phenotype (cluster 1, S4A, S4B Fig) and granulocytes (cluster 2, S4A, S4B Fig). Notably, monocyte/macrophages populations were enriched at 2 dpi, and a subpopulation of this cluster was associated with viral RNA (S4C, S4D Fig). Acknowledging the incompatibility between mouse Ace2 and SARS-CoV-2 spike, such association is likely the result of phagocytic activity, whose relevance for viral infection clearance will require investigations.

### Human lung epithelium signatures upon SARS-CoV-2 infection recapitulate features of COVID-19

We next examined the epithelial signatures of SARS-CoV-2 infection. Subclustering of the human epithelial compartment revealed subclusters of airway basal and secretory cells, alveolar type 1 (AT1) and type 2 (AT2) cells, and serous cells (Fig 5A). The dynamic changes of the epithelial compartment upon infection were consistent with COVID-19 human studies [2,5,3]. We noted a relative reduction of the AT2 compartment at 2 dpi (25.6%) compared to naïve mice (73.1%). The AT2 compartment was partially restored at 12 dpi (49.1%) (Fig 5B). The frequency of basal airway cells followed a reversed trend, indirectly reflecting the disruption of the AT2 compartment. The size of the AT1 compartment showed a relative increase at 2 dpi (9.89% compared to 8.00% in naïve fLX), consistent with the ability of AT2 cells to differentiate into AT1 upon lung damage [31] before returning to minimal relative levels as the AT2 compartment re-expanded following infection resolution.

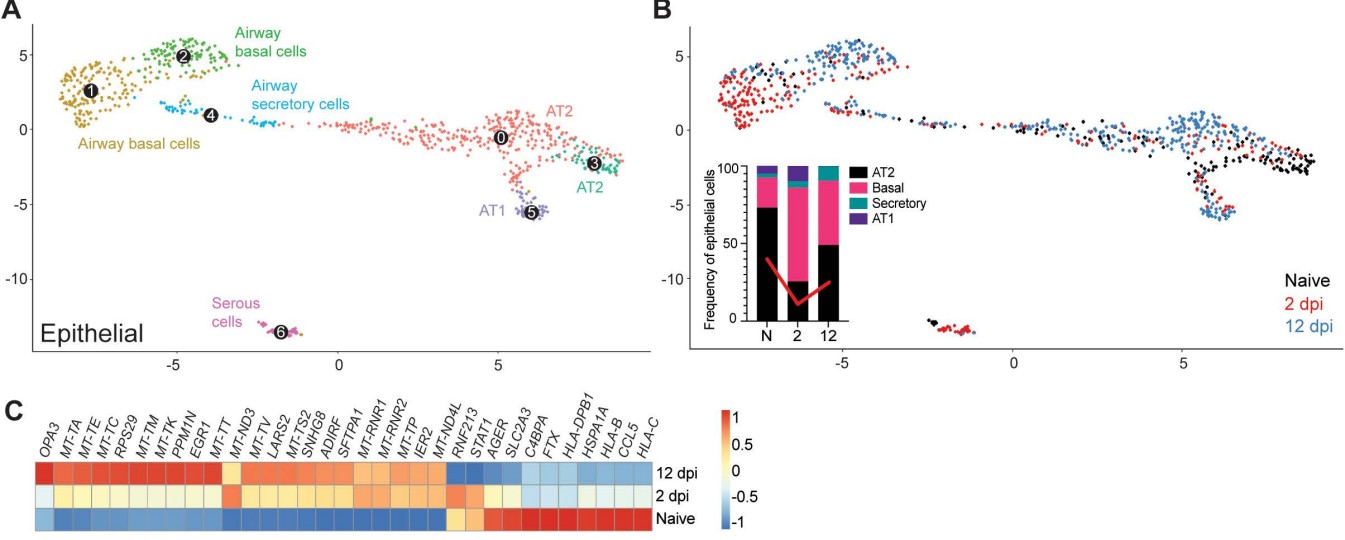

**Fig 5. Remodeling of the epithelial compartment of fLX during infection. (A)** UMAP plot subclustering of the human epithelial compartment in fLX across all time points analyzed by scRNA-seq. **(B)** Temporal annotation of the human epithelial subclusters on the UMAP plot: naïve (black), 2 dpi (red), 12 dpi (blue). The inset graph indicates the change in frequency of each compartment (AT2; black, Basal; pink, Secretory; teal, and AT1; purple) within the epithelial compartment. Red line indicates the change in the AT2 compartment over time. **(C)** Heatmap displaying the relative expression (from -1 to 1) of the top differentially regulated genes over the course of infection within the AT2 compartment.

The AT2 compartment displayed the most robust remodeling upon viral infection across the entire epithelial compartment, with 2 and 12 dpi AT2 subclusters showing distinctive transcriptomic signatures compared to naïve AT2 cells (Fig 5C). Interestingly, AT2 cells did not exhibit robust antiviral responses upon acute infection (Fig 5C). However, they displayed gene signatures previously reported during and after SARS-CoV-2 infection including, downregulation of MHC genes (*HLA-C, HLA-B, and HLA-DPB1)* and *CCL5* (a cytokine associated with immune recruitment during respiratory infection) [6], upregulation of immunomodulatory genes regulating inflammation and cell stress (*SFTPA1, EGR1*), and elevated expression of genes associated with mitochondrial dysfunctions and increased oxidative stress responses (i.e., *LARS2*). (Fig 5C). Activation of tissue repair mechanism (*SNHG8*) was also observed, further underscoring evidence of physiologically relevant AT2 response to viral infection (Fig 5C). The absence of detectable viral RNA in AT2 cells was in line with the lack of antiviral responses. Such a lack of infected AT2 cells in our scRNA-seq data is likely the reflection of their vulnerability and death during infection, given histological evidence of epithelial infection (Fig 1F, 1J) and the marked reduction of the AT2 compartment in fLX upon infection (Fig 5B).

Collectively, the epithelial compartment of fLX of BLT-L mice recapitulates several previously reported features of COVID-19 in humans, including loss of AT2 program, downregulation of immune genes and antigen presentation, and activation of tissue repair mechanisms.

## SARS-CoV-2 infection resolution is associated with the emergence of a viral RNA-enriched CD3-expressing macrophage-like cell subset

We first aimed to characterize the T-cell signatures to SARS-CoV-2 infection resolution, and with that, the features of our viral RNA-enriched, 2 dpi-specific, T-cell/ILC subcluster A. T-cell/ILC responses to infection were mainly limited to this specific viral RNA-associated subcluster (Fig 6A-6D), which segregated very distinctively from other CD3 + T-cell lineages, including canonical CD4+ and CD8 + T-cells (Subcluster 6; Fig 6A-6C). Using doublet discriminators and assessing viral RNA abundance, we determined that this subcluster was not a doublet and displayed low levels of *CD4* and *CD8* transcripts, suggesting a double-negative profile, and elevated levels of mitochondrial transcripts (S5A, S5B Fig). Most notably, it was uniquely defined by the expression of several transcripts known to be enriched in macrophages (*MARCO, TIMP1, LYZ*), fibroblasts (*MGP, CALD1, COL1A*), or both (*A2M, IFITM3*) but not in T-cells (Figs 6E and S5C). This subcluster also exhibited evidence of cytotoxic function through the expression of *GZMA, GZMB,* and *GNLY* transcripts, albeit expression was also detected in CD8 + T-cells and activated tissue-resident memory T-cells (TRM) (Subclusters 1,3,4, Fig 6E). Notably, human circulating monocytes can differentiate into CD3-expressing, double negative (CD4- and CD8-) macrophages. In mice, a similar subpopulation increases in the pleural space following intrapleural cavity injection of mycobacterium [32]. Given the proximity of this subcluster with myeloid lineages (Fig 3A, 3B) and evidence of expression of macrophage-enriched transcripts, we referred to this subcluster as CD3-expressing macrophage-like cells (MLC) [32]. MLC also exhibited upregulation of key gene pathways related to COVID-19, SARS-CoV-2 cell-intrinsic immune responses, cellular cytotoxicity (e.g., degranulation) and cell death, consistently with a role for this subset to serve as a robust primary responder to viral infection (Fig 6D). Notably, several evidence suggested active viral replication in MLC. Endogenous mRNA expression was shut down in MLC (S6A Fig), while ribosomal transcripts were overrepresented, suggesting high translational activity despite the low levels of mRNA (S6B Fig), consistent with the endogenous translational machinery being hijacked to support viral replication [33,34]. Biological processes related to mRNA metabolism were also downregulated along with many transcripts encoding for the basal mRNA transcriptional machinery (S6C, S6D Fig). Transcripts involved in cell responses to stress (S6E Fig), cellular macromolecule biosynthetic process (independent from ribosome activity; S6F Fig), and non-membrane-bounded organelle assembly (S6G Fig) were also specifically upregulated in MLC.

Of note, a recent human challenge study reported that self-resolving SARS-CoV-2 infection is associated with non-productive infection of human nasopharyngeal T-cells [14]. In contrast, in lung autopsy samples of fatal COVID-19

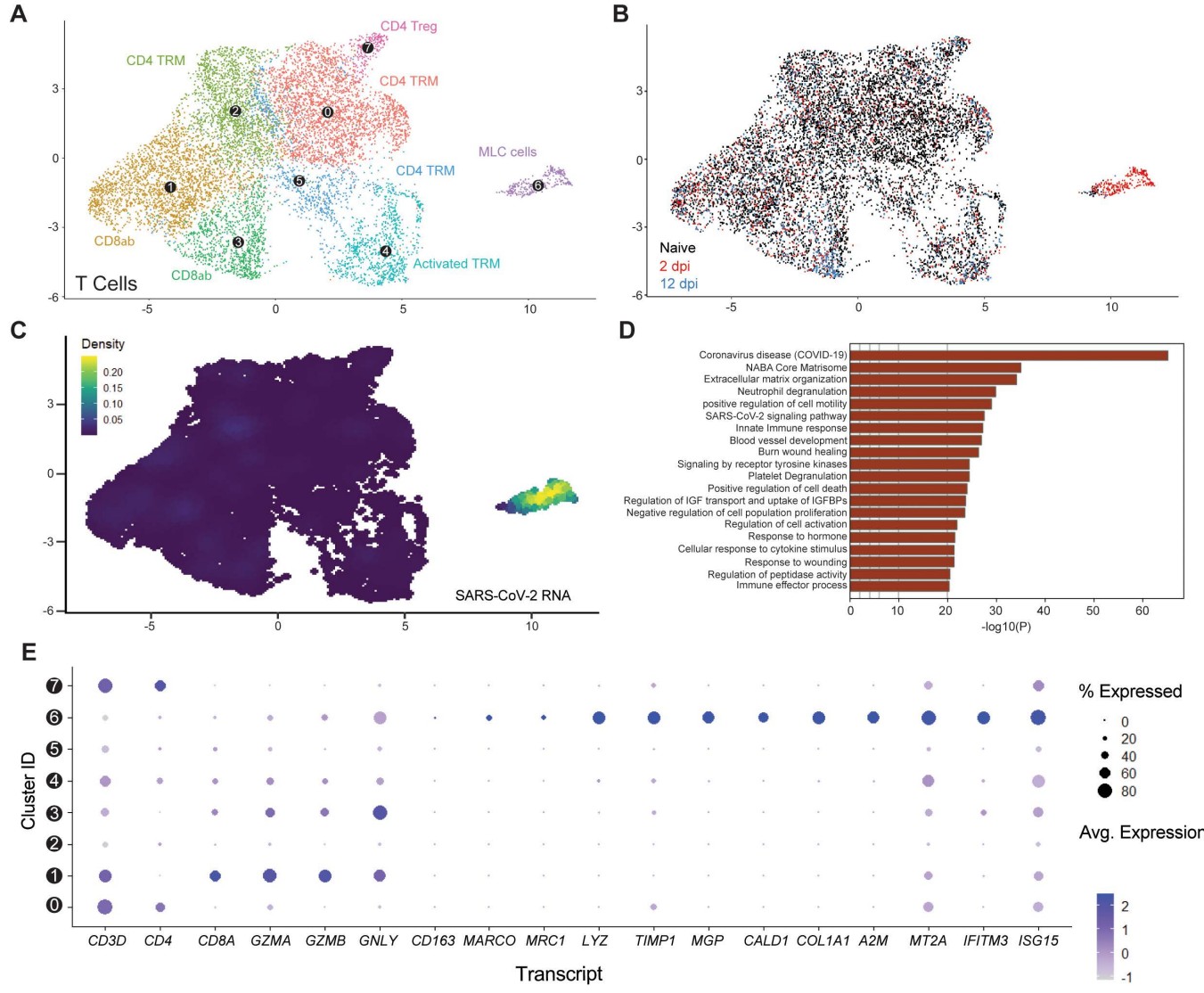

**Fig 6. Macrophage-like cells emerge upon acute infection and are enriched in viral RNA. (A)** UMAP plot showing the subclustering of the human T-cell compartment of fLX across all time points analyzed by scRNA-seq. **(B)** Temporal annotation of the human T-cell subclusters on the UMAP plot: naïve (black), 2 dpi (red),2 dpi (blue). **(C)** UMAP plot (all time points) showing the distribution of SARS-CoV-2 viral RNA transcripts in the T-cell compartment. **(D)** GO-term analysis showing the most highly upregulated signaling pathways in MLC (subcluster 6). **(E)** Relative expression of T cells, macrophage, mesenchymal and antiviral markers within each of the T-cell subclusters (From 0 to 7; as labeled in panel **A**).

cases, viral RNA was enriched in myeloid cells but was not detected in T-cells [35]. Collectively, our findings underscore that the emergence of viral RNA-enriched T cell populations displaying myeloid-like features in infected lung tissues is associated with effective SARS-CoV-2 infection resolution.

## Transient antiviral responses by a viral RNA-enriched inflammatory monocyte subset define lung myeloid responses driving SARS-CoV-2 infection resolution

Subclustering of myeloid lineages unveiled diverse subpopulations, including alveolar and macrophages (AM) and IM, various monocyte subsets, and one conventional DC subset (cDC) (Figs 7A and S7A-S7C). Across all time points,

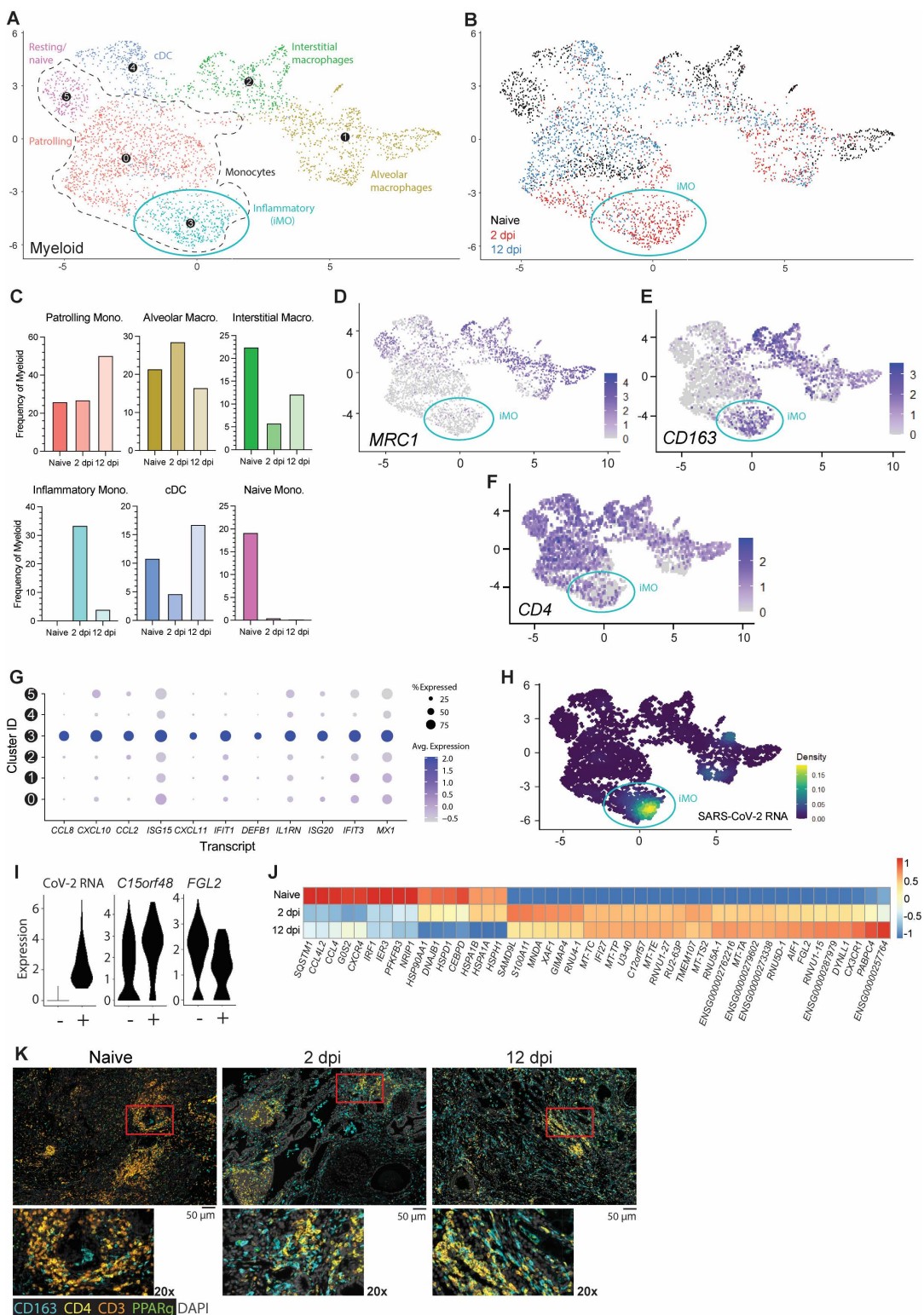

**Fig 7. CD163+extravascular inflammatory monocytes arise during acute viral infection and are predominant antiviral mediators. (A)** UMAP plot showing subclustering of the human myeloid compartment of fLX across all time points analyzed by scRNA-seq. **(B)** Temporal annotation of the myeloid subclusters on the UMAP plot: naïve (black), 2 dpi (red), 12 dpi (blue). iMO subcluster is indicated with a blue circle. **(C)** Frequency of each

myeloid subcluster by timepoint. **(D-F)** UMAP plot showing the expression of *MRC1* **(D)**, CD*163* **(E)**, and *CD4* **(F)** across myeloid subsets. **(G)** Relative expression of highly upregulated ISGs and inflammatory markers in each of the myeloid subclusters. **(H)** UMAP plot showing the distribution of SARS-CoV-2 viral RNA transcripts in the myeloid compartment across all time points analyzed by scRNA-seq. **(I)** Violin plots showing the expression level of differentially expressed genes between SARS-CoV-2 positive and negative iMO (subcluster 3). **(J)** Heatmap displaying the relative expression (from -1 to 1) of the top differentially regulated genes over the course of infection within the patrolling monocyte (0) compartment. **(K)** Multiplex fluorescent immuno-histochemistry of naïve fLX and fLX at 2 and 12 dpi. The images below represent 20x magnification of the indicated area (red). Images are representative of three independent fLX. CD163: teal, CD4: yellow, CD3: orange, PPARγ: green, Dapi: gray. Two representative images. Scale bar = 50 µM.

monocytes represented the largest sublineage, which was divided into three subgroups: naïve/resting, patrolling intra-vascular monocytes (PIM), and inflammatory monocytes (iMO). iMO and AM were the only two subclusters increasing in frequencies upon acute infection at the relative expense of cDC and IM (Fig 7B,7C). An increase in frequencies within these subclusters was associated with the emergence of distinctive iMO and AM cell populations with infection-induced transcriptomic signatures, underscoring a direct response of these subclusters against viral infection. Following infection resolution, most iMO had phased out and frequencies of alveolar macrophages were concomitantly reduced. Conjointly, the relative number of IM, cDCs and PIM increased through the recruitment of cell populations with distinctive transcriptomic identities from naïve and infection-associated cell populations (Fig 7B, blue).

Despite the moderate expansion of AM at 2 dpi, iMO were the most abundant myeloid lineage at 2 dpi (33.5%; Fig 7B, 7C) and uniquely defined among other monocyte subclusters through elevated expression of *VCAN, S100A8, CD14, CD163* and absence/minimal expression of *CD4, MARCO and CD206 (MRC1)* (Figs 7D-7F and S7D). iMO were also the leading mediators of antiviral responses across all other myeloid subclusters, as exemplified by the robust upregulated expression of interferon-stimulated genes (ISGs) and inflammatory cytokines (*CCL8, CXCL10, CCL2, ISG15, DEFB1, IL1RN, IFIT1, ISG20, IFIT3,* and *MX1*), as well as inflammatory markers such as *CD163* (Fig 7E,7G). While some ISGs (e.g., *ISG15, IFIT3, and MX1*) were expressed in other myeloid lineages at 2 dpi, their expression was markedly lower compared to iMO. Notably, upregulation of *CCL8*, *CXCL11* and *DEFB1* transcripts were the most exclusive to iMO (Fig 7G). iMO were also highly enriched in viral RNA, and corresponded to the previously referred 2 dpi-specific viral RNA-enriched subcluster B (Figs 3B, 4B, 7A,7B, and 7H), which also suggested an association between enrichment in viral RNA and potentiated antiviral responses. Concomitantly, we examined whether viral RNA-associated iMOs (CoV-iMOs) displayed a specific transcriptomic signature compared to iMOs (noCoV-iMOs) with undetectable viral RNA. Notably, only two genes correlated with the presence of viral RNA in iMOs: *FGL2 and C15orf48,* suggesting the absence of viral infection (abortive or productive) in these cells (Fig 7I). While *FGL2* was down-regulated in CoV-iMO compared to noCoV-iMOs, *C15orf48* was upregulated (Fig 7I). Soluble FGL2 exerts immunosuppressive functions (notably by inhibiting the NF-kB pathway) [36]. Conversely, the mitochondrial protein C15orf48 is positively regulated by NF-kB signaling [37] and has previously been implicated in severe COVID-19, acting as a positive regulator of inflammation [38]. A recurring feature of lung monocytes in fatal cases of COVID-19 is the expression of *IL-1β* [1,2,6–8], and inflammasome activation has been associated with the non-productive infection of monocytes [39]. However, no *IL-1β* expression or inflammasome activation was detected in CoV-iMOs despite enrichment in viral RNA.

In contrast to iMO, *CD4 + CD163- CD206(MRC1)*-PIM were detectable at all time points (**Subcluster 0**; Fig 7A-7C). 2 dpi-specific PIM harbored a distinctive, intermediate transcriptomic signature bridging naïve PIM and 2 dpi-specific iMO, and that was defined by the upregulation of specific interferon-stimulated genes (ISGs) such as *IFI27 and XAF1*, and pro-inflammatory genes (*S100A11*) (Fig 7J). At 12 dpi, PIM was the dominant subset over other monocyte subsets (50.3%; Fig 7C) and displayed a unique transcriptomic profile associated with anti-inflammation, tissue repair and cellular debris clearance mechanisms, notably through the upregulation of *FGL2, DYNLL1* and *CX3CR1*. At 12 dpi, expanded cDC and IM populations also elicited comparable transcriptomic signatures.

mIHC analysis supported our scRNA-seq findings. We observed a significant increase in CD4- CD3- CD163 + cells in fLX in the extravascular interstitium and alveolar spaces, mirroring the 2 dpi-specific expansion of CD4- CD163 + iMO and

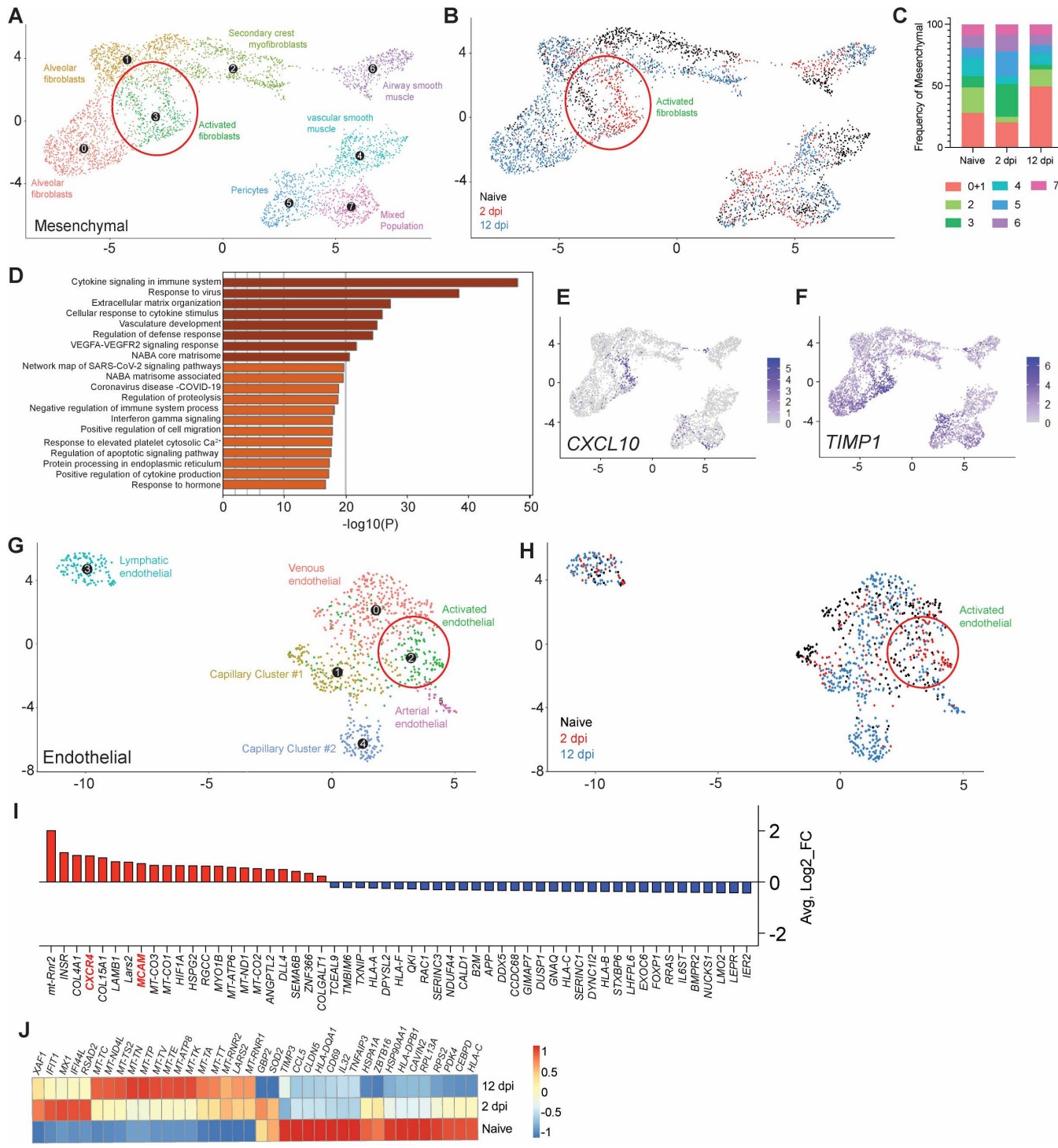

**Fig 8. Mesenchymal and endothelial signatures of infection resolution.** (A) UMAP plot subclustering of the human mesenchymal compartment in fLX across all time points analyzed by scRNA-seq. (B) Temporal annotation of the mesenchymal subclusters on the UMAP plot: naïve (black), 2 dpi (red), 12 dpi (blue). (C) Frequency of mesenchymal subclusters. (D) GO-term analysis showing cellular pathways linked to significantly upregulated genes within the activated fibroblasts subcluster. (E-F) UMAP plot showing *CXCL10* **(E)** and *TIMP1* **(F)** expression within all human mesenchymal subclusters and across all time points. Activated fibroblasts are indicated by a red circle in A and B. (G) UMAP plot subclustering of the human endothelial compartment of fLX across all time points analyzed by scRNA-seq. (H) Temporal annotation of the endothelial subclusters on the UMAP plot: naïve

(black), 2 dpi (red), 12 dpi (blue). (I) List of upregulated (red) and downregulated (blue) genes specific to the activated endothelial subcluster (Cluster 2) over the other endothelial subclusters (*p.adj* ≤ 0.05). (J) Heatmap displaying the relative expression (from -1 to 1) of the top differentially regulated genes over the course of infection within the venous endothelial compartment.

alveolar macrophages uncovered through scRNA-seq (Fig 7K). CD3- CD4 + CD163- at that time point could be associated with the 2 dpi-specific PIM population infiltrating the infected fLX. mIHC also recapitulated the reduction of CD3 + T-cells (Fig 7K) at 2 dpi. At 12 dpi, fLX were still enriched in CD4 + cells, likely reflecting PIM infiltration (Fig 7K). CD163 + cells also persisted in fLX at 12 dpi, which can be explained by the combined expansion of IM (CD4 + CD163+) and presence of AM (CD4- CD163+) at that time point as CD163 + iMO phase out (Fig 7K).

Collectively, our findings underscore a coordinated myeloid mobilization to infection resolution and tissue repair, with iMO concentrating viral materials and robust antiviral responses. As the viral materials are resolved, iMO populations dissipate, opening niches for other myeloid lineages to engage in a coordinated tissue repair process.

## Mesenchymal and endothelial signatures of infection resolution

We then examined the contribution of the mesenchymal and endothelial compartment in mediating effective infection resolution. Relative expansion of the mesenchymal compartment upon infection was mainly driven by an increase in fibroblast populations (Figs 3C, 8A and 8B). An increase in alveolar fibroblasts was observed at 12 dpi (49.6%; naïve = 28.1%), which was preceded by the emergence of a 2 dpi-specific cluster of activated fibroblasts (26.5%; naïve = 8.9%) (Fig 8A-8C), consistent with effective tissue repair in response to lung damage. Activated fibroblasts were the dominant mesenchymal population in mediating antiviral responses (Fig 8D), as displayed by robust upregulation of major pro-inflammatory mediators such as *CXCL10* and *TIMP1* (Fig 8F).

The endothelial compartment is an important modulator of immune recruitment through cytokine/chemokine signaling. Infected fLX showed the emergence of an activated, 2 dpi-specific endothelial cell cluster (Fig 8G, 8H), which was identified by increased expression of several transcripts involved in myeloid chemotaxis (Fig 8I), including *CXCR4* and *MCAM* [40,41]. Notably, venous endothelial cells also displayed upregulation of a panel of interferon-stimulated gene transcripts (*XAF1, IFIT1, MX1, IFI44L, RSAD2)* at 2 dpi (Fig 8J) and the downregulation of several transcripts coding for activation markers (*CD69*), MHC genes (*HLA-DPB1, HLA-C*) and major proteins regulating cellular metabolism and transcription (*HSP90AA1, RPL13A*), which was reflective of endothelial dysfunction and stress (Fig 8J) consistently with human patient reports and animal studies [42–45]. Collectively, our findings support the contribution of the mesenchymal and endothelium compartments in driving antiviral responses and myeloid chemotaxis, respectively, in driving infection resolution.

## Systemic depletion of CD4 + cells abrogates viral clearance in fLX

Many of our findings underscore a robust association between monocyte recruitment into fLX and SARS-CoV-2 infection resolution. This includes: 1) the recruitment of CD4 + PIM into infected fLX, 2) the monocyte nature of iMO and of their dominant antiviral responses, 3) the high enrichment in viral RNA of iMO during infection and 4) the endothelial-mediated myeloid chemotaxis signature at 2 dpi. These pieces of evidence also complement recent human findings which associate effective control of SARS-CoV-2 infection in the nasopharynx with monocyte recruitment [14]. Antibody-mediated depletions are commonly used to deplete specific hematopoietic subsets, and they are particularly amenable to HIS mouse model studies; further emphasizing the power of these *in vivo* platforms to mechanistically dissect immunological mechanisms in a human context. However, no anti-human CD14 antibodies have been well characterized for *in vivo* depletion of human monocytes. In contrast, anti-human CD4 antibodies have been. To experimentally validate the importance of the recruitment of circulating monocytes to drive infection resolution in fLX, we therefore performed systemic depletion of CD3 + , CD4 + , and CD8 + cells through the administration of OKT3, OKT4, or OKT8 depleting antibodies, respectively, via intraperitoneal injection of BLT-L mice both prior to and after infection (Fig 9A). Human monocytes, as well as some

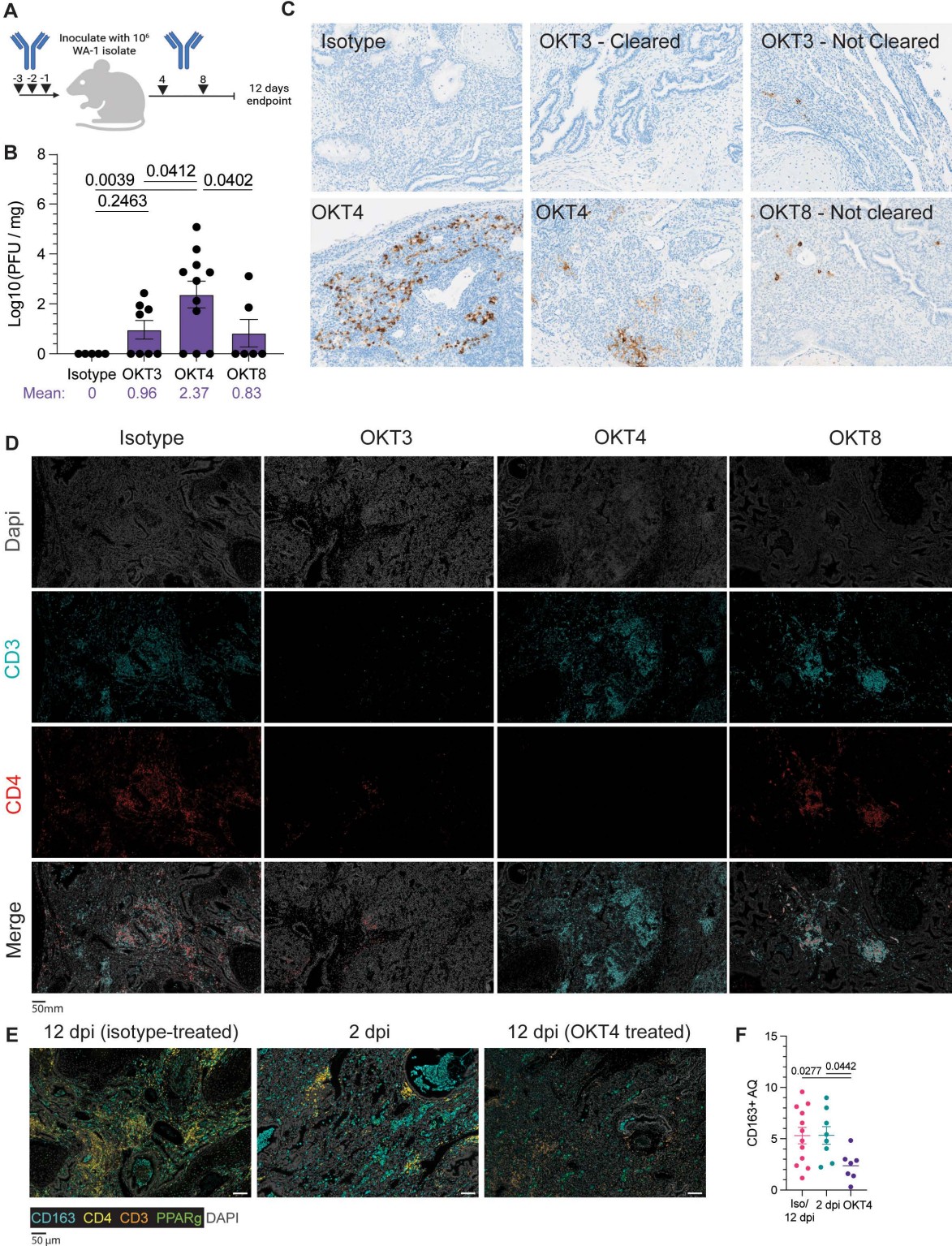

Fig 9. Systemic depletion of CD4+cells results in persistent infection of fLX. (A) BLT-L mice were administered 200 μg of anti-CD3e (OKT3), anti-CD4 (OKT4), anti-CD8 (OKT8) or IgG2a isotype. Created with BioRender.com. (B) Viral titer (log10(PFU/mg)) of fLX collected from BLT-L

mice treated with isotype, OKT3, OKT4, or OKT8 antibody at 12 dpi. (C) Immunohistochemistry for SARS-CoV-2 N protein of fLX collected from antibody-treated BLT-L mice at 12 dpi. Images are representative of two independent cohorts and donors. (D) Multiplex immunohistochemistry of fLX collected from antibody-treated BLT-L mice at 12 dpi. Dapi = gray, CD3 = teal, CD4 = red. Scale bar = 50 μm. Images are representative of two independent cohorts and donors. (E-F) Multiplex fluorescent immunohistochemistry (E) and CD163 + area quantification (AQ) analysis (F) of fLX following infection resolution (12 dpi, isotype-treated), or during acute (2 dpi) and persistent infection (12 dpi, OKT4-treated). Images are representative of two independent cohorts and donors. CD163: teal, CD4: yellow, CD3: orange, PPARγ: green, Dapi: gray. Scale bar = 50 μM.

macrophages and dendritic cells (DC) express CD4 [46,47], and the use of these three depleting antibodies will allow us to deconvolute the distinctive impact of T-cells and monocytes in driving infection resolution, as well as the potential contribution of mouse monocyte/macrophages in this process

Flow cytometry confirmed effective systemic depletion in the blood (S8A Fig). All animals were euthanized at 12 dpi to assess for the persistence of SARS-CoV-2 infection in fLX. While depletion of CD3 +, CD4 +, and CD8 + cells induced persistent infection in some or most of the animals, only CD4 + cell depletion (mean Log PFU/mg tissue = 2.37) resulted in a statistically significant defect in infection resolution compared to all other experimental conditions (isotype, OKT3, and OKT8-treated mice) (Fig 9B). The rate of productively infected fLX was also higher in CD4 + cell-depleted animals, with 72% of fLX (8/11) showing infection at 12 dpi compared to 50% (4/8) and 33% (2/6) in CD3- and CD8-depleted mice, respectively (Fig 9B). Notably, when examining the average viral titer (Log PFU/mg of tissue) of persistently infected fLX, the ones from CD4 + cell-depleted animals were about 21-fold higher compared to CD3 + cell-depleted fLX (mean Log10 PFU/mg tissue = 3.264 versus 1.930; S8B Fig), furthering that CD4 + cells have a more consequential impact on driving infection resolution than CD3 + cells. Anti-SARS-CoV-2 N IHC confirmed these findings and the superior defect of CD4 + cell-depleted animals to resolve infection (Fig 9C). Using multiplex fluorescent IHC (mIHC), we also validated the reduction in CD3 +, CD4+ and CD3+CD4- cells in fLX from CD3-, CD4+ and CD8 + cell-depleted animals respectively, compared to isotype-treated animals (Fig 9D). In the fLX of CD4 + cell-depleted mice, CD4 + cells depletion also associated with significant reduction of CD163 + cells at 12 dpi, underscoring the association between CD4 + infiltration into fLX and CD163 + cell recruitment and differentiation (Fig 9E, 9F).

Persistent infection in CD4 + cell-depleted animals was also associated with significant downregulation of MHC class I (Fig 10), a phenomenon we similarly observed in acutely infected fLX (2 dpi) (Fig 10) and that has been previously reported in cells with active SARS-CoV-2 replication [48,49]. This further emphasizes that depletion of CD4 + cells is associated with defective viral clearance mechanisms. These findings suggest that circulating CD4-expressing cells significantly mediate SARS-CoV-2 infection resolution in BLT-L mice.

## Defective monocyte recruitment is associated with systemic and local signatures of chronic infection

To further interrogate the impact of CD4 + depletion and monocyte recruitment on SARS-CoV-2 infection, we investigated fLX antiviral responses during acute and persistent infection by quantifying the concentration of 32 cytokines in the peripheral blood of naïve, acutely infected, persistently infected or recovered BLT-L mice. Systemic levels of human CCL2 and CCL3, major monocyte attractants, were elevated in CD4 + cell-depleted mice and comparable to those of acutely infected mice (Fig 11A, 11B). The maintenance of myeloid chemotaxis signals in persistently infected fLX further underlines the contribution of myeloid recruitment in SARS-CoV-2 infection resolution. Notably, among all cytokines and chemokines analyzed, CXCL10 was the only one displaying significantly increased serum levels in acutely infected mice (2 dpi) prior to returning to undetectable levels upon infection resolution (12 dpi) (Fig 11C). Several human subsets within fLX express *CXCL10* upon acute infection, including myeloid, mesenchymal and endothelial subsets (Fig 11D, 11E). However, the myeloid compartment was the major source of *CXCL10* among all human clusters at 2 dpi (Fig 11D, 11E) and this phenotype was dominantly driven by iMO (Figs 7G and 11E). Therefore, our findings suggest an association between circulating human CXCL10 during acute infection and the simultaneous differentiation of iMO. Notably, levels of circulating human

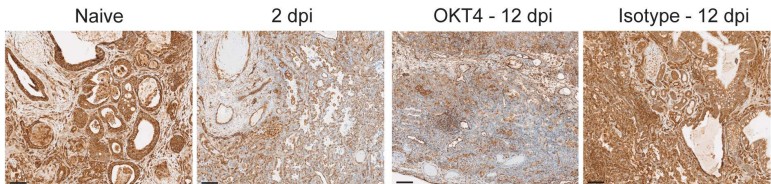

**Fig 10. MHC-I expression in fLX collected from naïve mice, antibody-treated mice and acutely infected mice.** MHC-I staining of fLX tissue sections collected from naive, isotype-treated (12 dpi), OKT4-treated (12 dpi), and acutely infected mice (2 dpi). Scale bar = 100 µM. Images are representative of two independent fLX for each experimental condition.

**Fig 11. CD4+cell depletion results in signatures of chronic infection.** (A-C) Cytokine quantification (**A**: CCL2, **B**: CCL3, **C**: CXCL10) in the serum of naïve, infected (2 dpi and 12 dpi isotype-treated or not) and OKT4-treated BLT-L mice (12 dpi). (D) Violin plot showing expression level of *CXCL10* per cell and within each human lineage. (E) UMAP plot showing *CXCL10* expression within all human lineages in naive fLX and at 2- and 12 dpi. Locations of the mesenchymal, endothelial, myeloid and iMO clusters at 2 dpi are indicated. (F) Quantification of CCL19 in the serum of naïve, infected (2 dpi and 12 dpi isotype-treated or not) and OKT4-treated BLT-L mice (12 dpi). (G) Histopathological scoring of fLX collected from isotype- and OKT4-treated BLT-L mice (12 dpi). Error bars indicate mean±Standard error of the mean. One-*way ANOVA, t-test. p-values are indicated on graphs.*

CXCL10 in persistently infected CD4 + cell-depleted mice were not statistically different than those of mice that resolved infection (12 dpi) (Fig 11C). These results emphasize a link between persistent infection and lack of effective CXCL10 responses, further strengthening a connection between monocyte recruitment, iMO antiviral responses and SARS-CoV-2 infection resolution.

Persistently-infected mice antiviral responses were also distinguishable from acutely infected (2 dpi) and isotype-treated (12 dpi) mice by elevated levels of CCL19 in serum (Fig 11F). CCL19 is a pro-inflammatory cytokine that has been linked with persistent viral replication and inflammation such as in the context of HIV-1 infection [50], underlining that persistent SARS-CoV-2 infection in BLT-L mice recapitulate key immunological features of chronic viral infection.

We have previously reported that fLX exhibit significant histopathological manifestations of disease upon SARS-CoV-2 infection in the absence of an engrafted human immune system [3], highlighting that tissue damage in fLX is virally induced. However, persistent infection did not result in any significant histopathological manifestations of disease in fLX, which strongly contrasts with acutely infected fLX (Figs 2H, and 11G and S1 Table). As minimal tissue damage is a hallmark of chronic viral infection [51], these findings further support evidence that CD4 + cell depletion promotes tissue remodeling processes underlying chronic infection and lasting viral persistence.

## Discussion

As our appreciation of the immunological differences between mice and humans continues to grow, humanized mouse models increasingly stand out as robust platforms to understand how viral pathogens interact with human tissues and the human immune system. These models are especially valuable when investigating tissue and mucosal immunity since such investigations remain impractical in human patients.

The SARS-CoV-2 pandemic has emphasized the need to increase our understanding of immune mechanisms that can drive protection against immunologically novel respiratory viruses. Mice engrafted with human immune systems and human lung tissues, which so far only include the BLT-L and HNFL mouse models, have emerged as valuable tools for such investigations [21,23–25], bridging the limitations of conventional animal models and the challenges associated with human studies. We previously reported that the HNFL mouse model can be leveraged to capture immunological signatures defining effective control of SARS-CoV-2 infection [23]. However, the enhanced myeloid reconstitution of the HNFL model rapidly inhibits viral replication in fLX, precluding our ability to study protective immunological mechanisms at play upon acute and potentially symptomatic infection. Using the BLT-L mouse model, a model that supports acute SARS-CoV-2 infection, the goal of our study was to capture human signatures of infection resolution in human lung tissues.

We found that BTL-L mice are able to effectively clear infectious viral particles following an early peak of viral replication in fLX. Infection resolution was associated with rapid mobilization of the human immune cells into fLX upon viral inoculation, which aligns with previous evidence that immunodeficient mice only engrafted with fLX are unable to clear infection [22,23]. Using scRNA-seq analysis, we identified a comprehensive network of novel factors associated with the resolution of SARS-CoV-2 infection (Fig 12). Our study identified three major hallmarks of this process: 1) The recruitment of iMO, which display a high level of viral RNA enrichment and are a major source of antiviral responses across all of the myeloid lineages, notably defined by CXCL10 expression; 2) The appearance of MLC cells, which are highly enriched in viral RNA, display evidence of viral infection, express CD3 and exhibit canonical macrophage features; And 3) The synergistic contribution of endothelial and mesenchymal cells in infection resolution via potentiating antiviral responses and myeloid chemotaxis, respectively. Consistent with monocyte infiltration into infected fLX being a dominant feature of our infection resolution model, systemic depletion of CD4 + cells, but not CD3 + cells, abrogated viral clearance. Persistent infection is associated with the lack of systemic CXCL10 responses, dominantly mediated by iMO, as well as with signatures of chronic infection, including systemic CCL19 expression and lack of fLX damage. By capturing these different signatures of infection resolution and persistent infection, our study paves the way for multiple lines of independent, mechanistic investigations. Our depletion experiment provides key evidence that our transcriptomic analysis can both guide and complement

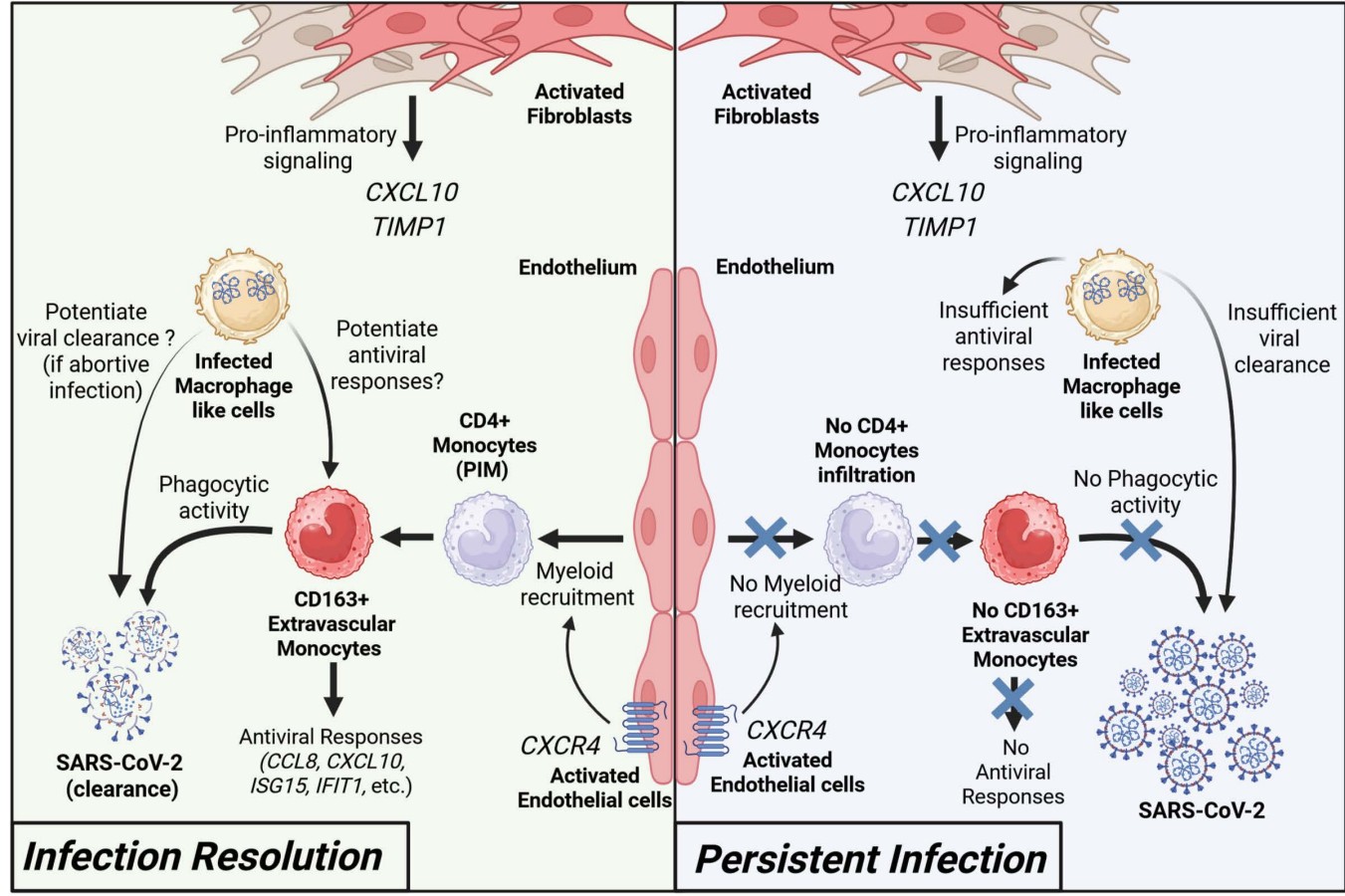

**Fig 12. Speculative model of SARS-CoV-2 infection resolution in human lung tissues.** During effective viral resolution and tissue repair (left panel), myeloid cells are recruited to the site of infection and differentiate into CD163 + inflammatory monocytes (iMO). iMO produce antiviral and inflammatory signals and exert phagocytic activity on viral particles. Additionally, CD3-expressing macrophage-like cells act as dead-end viral reservoirs, facilitating viral clearance, and elicit robust cytotoxic antiviral responses that overall potentiate infection resolution. In parallel, activated endothelial cells and fibroblasts contribute to the recruitment of myeloid cells and inflammatory response, respectively. Collectively, these events lead to viral clearance and infection resolution. In the context of persistent infection (left panel), absence of effective CD4 + monocyte recruitment prevents infection resolution. Created with BioRender.com.

the functional dissection of infection resolution mechanisms in BLTL mice, which we also demonstrate are suitable for such mechanistic studies.

Our findings support a recent human study reporting that protection from SARS-CoV-2 infection is associated with a rapid monocyte response in the nasal cavity and a decreased number of circulating monocytes [14]. Specifically, our work extends the findings of this human study beyond the limitations of the human model by exploring tissue-resident events, unraveling the identity of monocyte populations that extravasate into tissues, differentiate and mount tissue-resident immune responses to clear infection. By providing enhanced resolution on key protective immunological processes and mediators that human studies alone cannot capture, our work underscores how human and humanized mouse studies can effectively complement themselves for improving our understanding of human antiviral immunity.

To the best of our knowledge, this study represents the initial evidence of a direct role played by human lung extravascular inflammatory monocytes in the resolution of respiratory viral infections. Extravascular monocytes have been proposed to serve as immune sentinels through their position at the interface of the lung capillaries and alveoli [52,53].

However, although previous research reported that these cells can promote T-cell resident memory differentiation following viral infection [54], their direct antiviral functions have not been comprehensively examined so far. Four major features characterized these cells: 1) a dominant CD163-expressing population that emerges during acute infection before dissipating, 2) a major source of CXCL10 expression, 3) a high enrichment in viral RNA, and 4) the induction of robust antiviral responses.

Several lines of evidence support the monocyte nature of these cells. They display a close transcriptomic relationship with patrolling monocytes, which have been reported to give rise to transient, non-classical extravascular (including alveolar) monocytes in the mouse lung [53]. The iMO gene expression profile is also similar to those of human FCN1-monocytes recovered from the broncho-alveolar lavage of COVID-19 patients with acute respiratory disease syndrome [55]. While further studies are needed, our data collectively suggest CD4 + PIM infiltrating the infected fLX may differentiate into iMO to promote robust antiviral responses (Fig 12). Furthermore, our data show that some iMO are negative for viral RNA, suggesting that such differentiation is independent of an association with viral materials, although the presence of viral RNA could potentiate antiviral responses.

The fate of iMO following infection resolution also remains unclear and will have to be further deciphered. By 12 dpi and infection resolution, fLX display residual iMO and are enriched in CD4 + PIM and CD163 + CD206 + IM. CD206 + IM are involved in response to wounding and infection recovery [53] and CD163 + CD206 + monocyte-derived IM [55] have been shown to directly derive from extravasating, circulating CD14 + monocytes in another human immune system-engrafted mouse model [56]. Therefore, one hypothesis would be that the pool of CD163 + CD206 + IM is derived both from iMO and newly recruited circulating monocytes.

Our findings also reveal that different monocytic fates are likely associated with distinct clinical outcomes of SARS-CoV-2 infection. While our study emphasizes the protective role of monocytes in preventing severe COVID-19, monocytic lineages have also been associated with severe COVID-19 [2,5,3,55]. Excessive monocyte infiltration, macrophage inflammation and fibrotic response can lead to potentially fatal acute respiratory distress syndrome (ARDS) and fibrosis despite promoting infection resolution. A suspected driver of excessive myeloid inflammation is the ability of SARS-CoV-2 to trigger abortive infection of both monocytes and macrophages [39,57–59] through viral RNA replication and protein production, which results in inflammasome activation [39,60,61]. Consistently, viral RNA is enriched in inflammatory monocytes and macrophages in lung autopsy samples from fatal COVID-19 cases [5]. However, our findings suggest that the iMO transcriptomic antiviral signature is mostly independent of viral RNA, and that viral RNA-enriched iMO do not harbor signatures of productive infection or inflammasome activation. This suggests that monocyte antiviral and inflammatory responses, associated with favorable clinical outcomes are driven by phagocytic activity rather than infection (Fig 12). Of note, the abrogation of viral clearance via CD4 + cell depletion also support a negligeable contribution of mouse monocytes and macrophages (and of their phagocytic activity) to infection resolution in our mouse model.

Lung extravascular monocytes deriving from PIM exhibit a dynamic transition state and can differentiate into CD206-IM [53], which are primary responders to infection and important drivers of inflammation. However, through transcriptomic analysis, no *CD206-* macrophages were observed in infected fLX at any time point. The transient presence of viral RNA-enriched iMO in fLX, leading to the absence of iMO differentiation into *CD206-* IM, could therefore mitigate the risk of uncontrolled inflammation during infection resolution. This process could prevent severe tissue damage and excessive extravasation of circulating monocytes, which would otherwise differentiate into pro-fibrotic CD163 + CD206 + IM, a major driver of ARDS [55].

Understanding the cellular and molecular players defining the fate of iMO upon exposure to SARS-CoV-2 and how key regulatory crossroads in this subset may result in differential clinical outcomes is of particular interest. Compounding immune dysregulations, such as elevated inflammatory baseline and/or epigenetic imprinting related to innate immune training [62], may hinder PIM and/or iMO's ability to adequately regulate their inflammatory responses in a timely manner upon encountering viral materials or inflammatory clues. This could also favor extended cell survival (NF-κB is both

involved in inflammasome activation and cell survival [63]) and subsequent differentiation into inflammatory macrophages, fostering exacerbated inflammation. Our study also creates a mandate for probing the ubiquitous nature of iMO antiviral responses during other respiratory viral infections and investigating more comprehensively the antiviral functions of this novel subset beyond SARS-CoV-2 infection.

In parallel to iMO, we also identified MLC cells, a transient CD3-expressing cell population highly enriched in viral RNA and almost exclusively observed upon acute infection. They also uniquely exhibit expression of key macrophage-defining genes and cytotoxic markers. Albeit limited, there is evidence that circulating human monocytes can differentiate into CD3-expressing, monocyte-derived macrophages *in vitro* [32]. In a mouse model of pleural mycobacterium infection, such cells increase at the infection site during the acute stage prior to resolution to baseline levels [32]. MLC also share transcriptomic similarities with human γδ T-cell subsets previously described as bearing myeloid and cytotoxic functions [64]. The presence of mesenchymal markers in our MLC subpopulation suggests that CD3-expressing macrophages can also adopt experimental context-dependent phenotypes, underscoring the value of our experimental system to increase our understanding of these myeloid subpopulations. While CD8+T-cells and macrophages isolated from nasopharyngeal swabs have been identified to harbor SARS-CoV-2 RNA in human challenge studies [14], MLC did not meet the canonical transcriptomic signatures of these subsets, suggesting the existence of tissue-resident events driving specific cellular differentiation processes upon infection. Our transcriptomic findings also suggest that MLC, unlike iMO, support viral RNA translation and replication. Of note, SARS-CoV-2 infection of macrophages is abortive [39,57–59] and infection of T-cells, while productive, results in limited production of viral progenies [65]. Thereby, one could speculate that MLC infection can represent a dead-end for SARS-CoV-2 particles, potentiating both viral particles clearance from tissues and the induction of tissue-wide antiviral responses (Fig 12). Additional investigations will be required to better understand the identity, fate and functions of this cell subset in lung antiviral immunity, as well as its relative contribution to SARS-CoV-2 infection resolution relatively to iMO.

Beyond MLC, the overall contribution of CD3+cells in SARS-CoV-2 infection resolution remains elusive in our model. CD3+cell depletion resulted in a partial abrogation of infection resolution, with 50% of animals still capable of clearing infection. In addition, even when viral infection persisted in CD3+cell-depleted animals, viral titers in fLX remained significantly lower (21-fold lower) than in the fLX of persistently infected CD4+cell-depleted animals. Thereby, one could then speculate that CD3+cell depletion delays but does not abolish infection resolution. While T-cells showed no major transcriptomic responses to SARS-CoV-2 infection at 2 dpi, we cannot exclude that T-cell-mediated cytokine/chemokine signaling could hasten infection resolution. Alternatively, CD3+cell depletion could also prevent MLC (which express CD3) differentiation and the potential stimulatory impact of these cell's antiviral responses on the overall infection resolution process. Collectively, how CD3+depletion impacts MLC differentiation, as well as T-cells and iMO functions will need further investigation.

Patients suffering from immunodeficiencies can experience persistent SARS-CoV-2 replication in the respiratory tract for over 100 days [66–68]. Persistent infection in immunodeficient individuals has a strong public health significance, since extended periods of replication in a single host have been postulated to favor the emergence of variants of concern [66,69]. Persistent SARS-CoV-2 infection in BLT-L mice is associated with three major features: [1] persistent CCL2 and CCL3 production, [2] absence of lung histopathological manifestation of disease and [3] elevated serum concentration of CCL19. Notably, CCL19 has been associated with persistent inflammation in untreated HIV-1 patients [50] as well as with immune tolerance mechanisms [70]. Furthermore, while this study, as well as previous reports [22,23], support evidence of an association between acute SARS-CoV-2 replication and histopathological manifestations of disease in fLX regardless of the co-engraftment of a human immune system, the absence of such manifestations in infected fLX from CD4+cell-depleted suggests a potential remodeling of the epithelial compartment toward viral tolerance. Therefore, the ability of the OKT4-treated BLT-L mice to recapitulate productive and persistent SARS-CoV-2 infection opens avenues to comprehensively dissect viral, epithelial and hematopoietic adaptations favoring viral persistence in a human tissue

context and viral genetic drift. This model also offers promises to explore how persistent viral replication and/or the maintenance of specific inflammatory signals can drive the development of epithelial and mesenchymal-associated pathologies in the lung, including pulmonary fibrosis.

There are inherent limitations associated with the BLT-L mouse model. First, direct viral inoculation into fLX potentially bypasses key immune checkpoints in the upper respiratory tract, which are not recapitulated in our model. Second, the hematopoietic reconstitution and functions of the BLT-L mouse model remain imperfect, notably through the underrepresentation of specific hematopoietic subsets, including dendritic cells and granulocytes, and limited B-cell responses – which may ultimately bias the dynamics of viral clearance. While the enhanced myeloid compartment of the HFNL mouse model addresses some of these limitations, it also drives a more rapid clearance process that manifests by the lack of detectable acute infection. Ultimately, this illustrates how the specificity of each humanized mouse model can be leveraged to address specific questions. Despite these limitations, BLT-L mice recapitulated human lung responses to SARS-CoV-2 infection through AT2 loss of programming, fibroblast and endothelial activation, and effective clearance of infection through robust hematopoietic responses. This mouse model also recapitulated hallmarks of chronic infection in patients. Our work demonstrates the potential of the BLT-L mouse model to uncover naïve immune mechanisms and mediators governing the effective resolution of lung infection by SARS-CoV-2 and open avenues for a comprehensive examination of such processes against other viral respiratory infections, which may pave the way toward innovative immunotherapy strategies against these diseases.

## Materials and methods

Detailed descriptions of the materials and methods used for this study are available in Supporting Information.

### Ethics statement

All experiments in this study, including those conducted in BSL-3, were approved by an institutional biosafety committee. Animal experiments described in this study were performed in accordance with protocols that were reviewed and approved by the Institutional Animal Care and Use and Committee of the Ragon Institution and Boston University. All mice were maintained in facilities accredited by the Association for the Assessment and Accreditation of Laboratory Animal Care (AAALAC). All replication-competent SARS-CoV-2 experiments were performed in a biosafety level 3 laboratory (BSL-3) at the Boston University National Emerging Infectious Diseases Laboratories (NEIDL).

### Mouse strains and sex as biological variable

Female NOD.Cg.-$Prkdc^{Scid}Il2rg^{tm1Wjl}$/SzJ (NSG) mice were obtained from the Jackson Laboratory, catalog number 005557. NSG mice were maintained by the Ragon Institute Human Immune System Mouse core prior to engraftment and shipment to the NEIDL, Boston University. In our study, only female mice were engrafted with human fetal tissues because of their ability to support higher levels of engraftment than males. As we investigated human tissue responses to infection and only leveraged mice as xenorecipients, the sex of the animals does represent a critical variable for our study.

### Human fetal tissues

De-identified human fetal tissues were procured from Advanced Bioscience Resources (Alameda, CA, USA).

### Generation of BLT-L mice

BLT-L mice were generated via irradiation of female NOD.Cg.-$Prkdc^{Scid}Il2rg^{tm1Wjl}$/SzJ mice (NSG mice; Jackson Laboratory #005557) prior to implantation of human fetal thymic and fetal liver tissue (Advanced Bioscience Resources) under the murine kidney capsule. A single piece of homologous human fetal lung tissue was implanted into the subcutaneous dorsal pocket of female NSG mice. Post-implantation, mice intravenously received 1x10$^5$ homologous CD34 + cells. Human

immune reconstitution was determined by flow cytometry 19–20 weeks post-implantation. Three distinct human donors were used for this study across multiple experiments. Three donors were used for viral titer quantification experiments. One donor was used for scRNA-seq experiments. Two donors were used to conduct depletion assays and independently validate findings.

## Mouse inoculation with SARS-CoV-2

BLT-L mice were challenged 23–28 weeks post-engraftment. Mice were anesthetized with 1–3% isoflurane prior to inoculation via subcutaneous, intra-fetal lung xenograft (intra-fLX) injection with $10^6$ plaque forming unites (PFU) of SARS-CoV-2 WA-1 isolate in 50 μL of sterile 1X PBS. All non-infected animals in this study, referred to as naïve, were inoculated with 50 μL of sterile 1X PBS. Mice were euthanized at 2-, 6-, and 12-days post inoculation.

## In vivo antibody depletion

BLT-L mice were administered 200 mg of anti-CD3e (OKT3) (BioxCell; cat. # BE0001–2), anti-CD4 (OKT4) (BioxCell; cat. # BE0003–2), anti-CD8 (OKT8) (BioxCell; cat. # BE0004–2), or isotype IgG2a (Thermofisher; cat # 02–6200) antibody 3-, 2-, and 1-day prior to inoculation and 4- and 8-days post inoculation with $1\times10^6$ PFU SARS-CoV-2 WA-1.

## Quantification and statistical analysis

For histopathological score and viral load/titer comparisons a Kruskla-Wallis, non-parametric one-way ANOVA with Benjamini, Krieger, and Yekutieli correction for multiple comparisons was applied given the non-continuous nature of the data. For cytokine data, an Ordinary one-way ANOVA with uncorrected Fishers LSD was used as the data was collected from different time points, treatment conditions, and cohorts. A Kruskla-Wallis, non-parametric one-way ANOVA with an uncorrected Dunn's test was applied for CD163+Area quantification (AQ) due to the independent comparisons between the samples. All statistical tests and graphical depictions of results were performed using GraphPad Prism version 9.0.1 software (GraphPad Software, La Jolla, CA). For all tests, $p \leq 0.05$ was considered statistically significant. Statistical significance on figures and supplemental figures is labeled with p-values and non-significant values are labeled with n.s. or left unlabeled.

## Supporting information

**S1 Fig. BLT-L mice are susceptible to SARS-CoV-2 infection. (A-C)** Transmission electron microscopy (TEM) of fLX tissue sections extracted from BLT-L mice at 2 dpi, illustrating virus particles at the cell surface as indicated by the blue arrows and blue dotted line **(A)** and viral particles in AT2 cells as evident by the presence of lamellar bodies **(B,C)**. Scale bars are indicated in the images.
(TIF)

**S2 Fig. Neutralizing antibody-independent Spike selective pressure within fLX. (A)** Neutralizing efficacy (ND50 values) of serum extracted from naïve or infected BLT-L mice (2 and 12 dpi). Assay was performed using VeroE6 cells and a recombinant WA-1 SARS-CoV-2 virus expressing NanoLuc. **(B)** Titration of the neutralizing activity of an anti-RBD antibody (serving as positive control) against WA-1 SARS-CoV-2 virus expressing NanoLuc. **(C-D)** SARS-CoV-2 virus isolated from 2 dpi fLX was deep sequenced and assessed for mutations relative to the inoculation (WA-1) strain. **C Upper panel**: Pie chart depicting the percentage of fLX with virus containing 216KLRS insertion and R245H non-synonymous mutation. **C Lower Panel**: Schematic representation of SARS-CoV-2 spike protein highlighting the location of mutations. *Not to scale.* **D:** Pie chart depicting the percentage of fLX with virus that showed signs of co-evolution of 216KLRS and R245H.
(TIF)

                                                                          

**S3 Fig. Single-cell RNA sequencing analysis of fLX upon SARS-CoV-2 infection. (A-E)** Single-cell RNA sequencing was performed on naïve fLX and infected fLX (at 2- and 12 dpi). Sequencing reads were aligned to a combined human, mouse and SARS-CoV-2 viral genome. **(A)** UMAP plot clustering on all cell (human and mouse) populations detected. **(B)** Expression of mouse (top) and human *GAPDH* (bottom) in all clusters. **(C)** A human score was applied to each cluster to identify human clusters (above dotted line) and mouse clusters (below dotted line; gray zone). Mouse clusters were removed from downstream analysis. **(D)** Cell scoring system applied to human cell clusters to classify each cluster. **(E)** UMAP clustering of human cells separated by time point (naïve: left, 2 dpi: center, and 12 dpi: right). **(F)** Flow cytometric analysis showing frequency of hCD45+, hCD3+, hCD4+, and hCD8+ cells among PBMCs extracted from the blood of naïve and infected BLT-L mice at 2-, 6-, and 12 dpi.
(TIF)

**S4 Fig. Single-cell RNA sequencing analysis of mouse cells within fLX.** Single-cell RNA sequencing was performed on naïve fLX and infected fLX (at 2- and 12 dpi). Sequencing reads were aligned to a combined human, mouse and SARS-CoV-2 viral genome. **(A)** UMAP plot clustering of the mouse cell compartment of naïve fLX and fLX at 2 and 12 dpi. **(B)** UMAP plot showing expression of select genes used for annotating mouse clusters. **(C)** Temporal annotation of the mouse clusters on the UMAP plot: naïve (black), 2 dpi (red), 12 dpi (blue). **(D)** UMAP plot showing the distribution of SARS-CoV-2 viral RNA transcripts by density across all mouse cell clusters and time points analyzed by scRNA-seq.
(TIF)

**S5 Fig. Gene signature of macrophage-like cells. (A)** Mitochondrial gene expression among the different T cell sub-clusters. **(B)** Expression of T-cell-associated genes among the different T-cell sub-clusters. **(C)** Expression of myeloid/macrophage-associated genes among the different T-cell sub-clusters.
(TIF)

**S6 Fig. Evidence of viral infection in macrophage-like cells. (A)** Number of total RNA counts in T cell clusters. **(B)** Ribosomal score for each T cell cluster. **(C)** Pathway analysis of significantly downregulated genes in macrophage-like cell cluster (cluster 6). **(D)** Expression of select genes related to mRNA translation across T cell clusters. **(E)** Expression of select genes related to cellular stress responses across T cell clusters. **(F)** Expression of select genes related to macromolecule biosynthesis in T cell clusters. **(G)** Expression of select genes related to non-membrane-bounded organelle assembly.
(TIF)

**S7 Fig. Gene signature of myeloid and endothelial sub-clusters. (A-C)** Violin plot displaying monocyte **(A)**, macrophage **(B)** and dendritic cell (cDC) **(C)** gene signature score for each myeloid sub-cluster. **(D)** UMAP plots representing the expression of several myeloid markers that were employed to further define the identity of the different myeloid sub-clusters.
(TIF)

**S8 Fig. Assessment of systemic depletion efficiency and analysis of viral titers in mice with abrogated viral clearance mechanisms. (A)** Flow cytometric analysis showing the frequency of hCD45+, hCD3+, hCD3+hCD4+, and hCD3+hCD8+ cells among PBMCs extracted from the blood of BLT-L mice treated with an isotype antibody, OKT3, OKT4 or OKT8 antibody. Error bars represent mean±SEM. *One-way ANOVA. P-values are indicated on the graphs.* **(B)** Viral titer (log(PFU/mg)) of CD3+ cell (OKT3) and CD4+ cell-depleted (OKT4) fLX that were positive for viral infection. *Unpaired, non-parametric T-test. P-value is indicated on the graph.*
(TIF)

**S1 Table. Histopathological scoring of fLX.** Detailed histopathological scoring for each fLX analyzed in this study.
(DOCX)

**S2 Table. SARS-CoV-2 mutations in infected fLX.** SARS-CoV-2 mutations identified among viral reads isolated from fLX at 2 dpi and from our concentrated viral stock (i.e., WA1 positive stock used to inoculate fLX), as compared to the Wuhan-1 consensus sequence.
(DOCX)

**S1 Materials and Methods. Detailed description of the resources and experimental procedures used in this study.**
(DOCX)

## Acknowledgments

We thank the Evans Center for Interdisciplinary Biomedical Research at Boston University Chobanian and Avedisian School of Medicine for their support of the Affinity Research Collaborative on 'Respiratory Viruses: A Focus on COVID19'. We thank Ronald B. Corley, NEIDL director at the time of this study and MassCPR award recipient, for his constant encouragement and support. We thank the Boston University Animal Science Center, the Ragon Institute Human Immune System Mouse core, the Single Cell Sequencing Core, the Microarray and Sequencing Core, and the Flow Cytometry Core at the Boston University Chobanian and Avedisian School of Medicine, the SFR F. Bonamy Bioinformatics Core Facility, and all the NEIDL animal core staff for their outstanding support. We also thank all the Douam, Balazs, Connor, and Crossland lab members, NEIDL members, and members of the Department of Virology, Immunology, and Microbiology and Pathology at Boston University for their constant support and advice.. We used Grammarly (Grammarly Inc.) to assist with grammar correction and sentence structure optimization during manuscript preparation.

## Author contributions

**Conceptualization:** Devin Kenney, Christelle Harly, Alejandro B. Balazs, Florian Douam.

**Data curation:** Devin Kenney, Anna E. Tseng, Jacquelyn Turcinovic, Maria Ericsson, Nicholas A. Crossland, Christelle Harly, Florian Douam.

**Formal analysis:** Devin Kenney, Anna E. Tseng, Jacquelyn Turcinovic, Adam D. Nitido, Paige Montanaro, Maria Ericsson, Vladimir Vrbanac, Nicholas A. Crossland, Christelle Harly, Florian Douam.

**Funding acquisition:** Nicholas A. Crossland, Alejandro B. Balazs, Florian Douam.

**Investigation:** Devin Kenney, Aoife K. O'Connell, Anna E. Tseng, Jacquelyn Turcinovic, Maegan L. Sheehan, Adam D. Nitido, Paige Montanaro, Hans P. Gertje, Vladimir Vrbanac, Christelle Harly, Florian Douam.

**Methodology:** Devin Kenney, Anna E. Tseng, Jacquelyn Turcinovic, Maria Ericsson, Vladimir Vrbanac, Nicholas A. Crossland, Christelle Harly, Alejandro B. Balazs, Florian Douam.

**Project administration:** Nicholas A. Crossland, Christelle Harly, Alejandro B. Balazs, Florian Douam.

**Resources:** John H. Connor, Vladimir Vrbanac, Nicholas A. Crossland, Christelle Harly, Alejandro B. Balazs, Florian Douam.

**Software:** Jacquelyn Turcinovic, Christelle Harly.

**Supervision:** John H. Connor, Nicholas A. Crossland, Christelle Harly, Alejandro B. Balazs, Florian Douam.

**Validation:** Nicholas A. Crossland, Christelle Harly, Florian Douam.

**Visualization:** Devin Kenney, Anna E. Tseng, Jacquelyn Turcinovic, Nicholas A. Crossland, Christelle Harly, Florian Douam.

**Writing – original draft:** Devin Kenney, Florian Douam.

**Writing – review & editing:** Devin Kenney, John H. Connor, Vladimir Vrbanac, Nicholas A. Crossland, Christelle Harly, Alejandro B. Balazs, Florian Douam.

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
