## [Decision Letter · Decision Letter 0]

22 Apr 2025

PPATHOGENS-D-24-02817

Immune Signatures of SARS-CoV-2 Infection Resolution in Human Lung Tissues

PLOS Pathogens

Dear Dr. Douam,

Thank you for submitting your manuscript to PLOS Pathogens. After careful consideration, we feel that it has merit but does not fully meet PLOS Pathogens's publication criteria as it currently stands. Therefore, we invite you to submit a revised version of the manuscript that addresses the points raised during the review process. The manuscript is being returned with three reviews. Each reviewer identified a number of specific concerns, and these concerns should be addressed in a revised manuscript. Major areas of concern identified by multiple reviewers include the following:

1. R1 and R3 raised concerns with cell populations in the scRNAseq data. 

2. All reviewers identified issues with the experimental design that require clarification and/or modification.

3. R2 and R3 raised important concerns regarding conclusions based on scRNAseq data in the absence of functional validation; this is particularly true for the T cell population with myeloid-like features.

4. Multiple reviewers raised issues regarding the evidence presented to support chronic infection. 

Please submit your revised manuscript within 60 days Jun 21 2025 11:59PM. If you will need more time than this to complete your revisions, please reply to this message or contact the journal office at plospathogens@plos.org. Please include the following items when submitting your revised manuscript:

We look forward to receiving your revised manuscript.

Kind regards,

Thomas E. Morrison

Academic Editor

PLOS Pathogens

Alexander Gorbalenya

Section Editor

PLOS Pathogens

Sumita Bhaduri-McIntosh

Editor-in-Chief

PLOS Pathogens

orcid.org/0000-0003-2946-9497

Michael Malim

Editor-in-Chief

PLOS Pathogens

orcid.org/0000-0002-7699-2064

**Journal Requirements:**

https://journals.plos.org/plospathogens/s/submission-guidelines#loc-parts-of-a-submission

3) We noticed that you used the phrase 'data not shown' in the manuscript. We do not allow these references, as the PLOS data access policy requires that all data be either published with the manuscript or made available in a publicly accessible database. Please amend the supplementary material to include the referenced data or remove the references.

4) We do not publish any copyright or trademark symbols that usually accompany proprietary names, eg ©,  ®, or TM  (e.g. next to drug or reagent names). Therefore please remove all instances of trademark/copyright symbols throughout the text, including:

- TM on page: 26.

5) Please upload all main figures as separate Figure files in .tif or .eps format. For more information about how to convert and format your figure files please see our guidelines: 

6) We have noticed that you have uploaded Supporting Information files, but you have not included a list of legends. Please add a full list of legends for your Supporting Information files after the references list.

7) Some material included in your submission may be copyrighted. According to PLOSu2019s copyright policy, authors who use figures or other material (e.g., graphics, clipart, maps) from another author or copyright holder must demonstrate or obtain permission to publish this material under the Creative Commons Attribution 4.0 International (CC BY 4.0) License used by PLOS journals. Please closely review the details of PLOSu2019s copyright requirements here: PLOS Licenses and Copyright. If you need to request permissions from a copyright holder, you may use PLOS's Copyright Content Permission form.

Potential Copyright Issues:

- Figures 1 and 9. Please confirm whether you drew the images / clip-art within the figure panels by hand. If you did not draw the images, please provide (a) a link to the source of the images or icons and their license / terms of use; or (b) written permission from the copyright holder to publish the images or icons under our CC BY 4.0 license. Alternatively, you may replace the images with open source alternatives. See these open source resources you may use to replace images / clip-art:

8) Please amend your detailed Financial Disclosure statement. This is published with the article. It must therefore be completed in full sentences and contain the exact wording you wish to be published. Please ensure that the funders and grant numbers match between the Financial Disclosure field and the Funding Information tab in your submission form. Note that the funders must be provided in the same order in both places as well.

**Reviewers' Comments:**

Reviewer's Responses to Questions

**Part I - Summary**

Reviewer #1: The manuscript “Immune Signatures of SARS-CoV-2 Infection Resolution in Human Lung Tissues” by Kenney et al. provides a very comprehensive characterization of immune signatures, using single cell transcriptomics, of resolution of SARS-CoV-2 infection in a human lung xenograft mouse model. The work is a follow-up of the observation by the same group that reconstitution of the human immune system resulted in a transient SARS-CoV-2 infection, rather than prolonged infection observed in lung-only models. The authors identify several cell populations that play a role in this resolution, and provide experimental data supporting the role of CD4+ cells. Overall, the experiments are well designed and the data support the conclusions. However some clarification is needed for interpretation.

Reviewer #2: In this manuscript entitled “Immune Signatures of SARS-CoV-2 Infection Resolution in Human Lung Tissues” by Kenney et al. mapped the immunological events during SARS-CoV-2 infection in the lung in a unique mouse model with the mice co-engrafted with a genetically matched human immune system and fetal lung xenografts. In this novel mouse model SARS-CoV-2 infection is rapidly cleared in the fetal lung tissue with rapid recovery of the histopathologic changes seen in SARS-CoV-2 infection. Surprisingly despite having no evidence of a humoral response (due to the limitations of the humanized mouse model system) they still showed evidence of viral evolution/adaptation. Interestingly these histopathologic changes and loss of the AT2 program seen in the lung epithelial cells in human COVID-19 patients was also recapitulated in this model platform. The authors note an alteration and enrichment of a T-cell subcluster that when analyzed more carefully displayed low levels of CD4 and CD8 with high-level expression of macrophage markers which they dubbed double-negative T-cells. Of great interest was the finding that depletion of CD4+ cells led to the development of a prolonged infection with multiple cellular and immune alterations. Moreover, the study shows the important role of infiltrating monocytes in driving resolution of SARS-CoV-2 infection. This is a highly relevant topic, as the delineation of SARS-CoV-2 induced injury and repair is important in understanding lung-pathogen responses and the protective role of the myeloid compartment represents an improved understanding of SARS-CoV-2 infection. Overall, this is manuscript is a robust work with a complex model platform that addresses several open areas in the field however, several issues need to be addressed before the manuscript can be considered for publication:

Major comments

1. Histopathologic changes seen in the human lung parenchyma are somewhat hard to interpret as no higher power H&E or trichrome images were shown (Fig 2). The histopathologic score which is a unique scoring system developed by the investigators in a prior published manuscript should still be elaborated in this manuscript and in the text as it is not a gold-standard analysis technique and therefore the interpretation of the data is not clear as it is not clear what it being quantified (Fig.2H). It would also add value (if shown in a table in the supplement) what is being quantified and what components of the score system is driving the values seen. Do the investigators have any comment on the differences seen in the score seen in their prior Cell Reports paper and the values seen in this report?

2. The investigators see a minor population of chondrocytes present in their scRNAseq analysis. Typical lower airway or even lung parenchyma analysises usually do not pull this population of cells due to cartilage only being present in the trachea and upper airways. Is this population present due to the portions/locations of lung tissue that were collected or due to the fetal nature of the human lung tissue leveraged? Do the authors believe that initial fetal lung tissue collection (location) would impact and/or alter the findings seen? And is this at all reflected in the heterogeneity seen in the cell composition and responses?

3. In figure 5 no club cells were noted. Is this not present in fLX or somehow lost in the analysis? Alterations in this compartment have been previously noted in earlier SARS-CoV-2 studies.

4. The authors identify a novel T-cell subcluster that when analyzed more carefully displayed low levels of CD4 and CD8 with high-level expression of macrophage markers which they dubbed double-negative T-cells. Given the unique nature of the model system and to ensure that this is not an artifact of the BLT flX model platform better linking of this population of cells to prior reports in human and in human viral infection would add greater strength and context to the manuscript.

5. Figure 7K IF staining used to validate the enriched population seen by scRNAseq is hard to visualize due to the magnification. High mag images and report on the number of images this is representative would add strength to the data.

6. The depletion of CD4 led to the persistence of SARS-CoV-2 infection which is a profound finding and the claim of chronic infection. Were later timepoints examined to demonstrate persistent infection? This is a really interesting model and no comment is made on the impacts of prolonged or “chronic infection” on the epithelial and mesenchymal/stromal compartments. As similar to that reported in prolonged human infections do the authors see the development of “pathological” fibroblasts and continued or dysregulated epithelial responses?

7. While some limitations were noted with the BLT-L mouse model further comparison with prior humanized lung and immune system mice would provide greater contrast and comparison with the similarities and differences with such models and would help to better put the author’s work in context.

Reviewer #3: (No Response)

**Part II – Major Issues: Key Experiments Required for Acceptance**

Reviewer #1: The authors have previously established and characterized this model in their lab, which is already published. However, some key aspects should be revisited here in order to be able to interpret the data:

For example, NSG mice have been shown to be susceptible to infection of murine tissue by SARS-CoV-2. This could result in systemic effect that could (in part) influence responses in the lung xenografts. In fact, the authors also mention the presence of a minor population of murine cells in the scRNA-seq data, but excluded this from analyses. It would be important to specify what cell types were mostly associated, are these immune cells, or for example, endothelial cells involved in vascularization.

The level of humanization (hCD45+) can vary between animals/donors etc. In Figure S3F, this ranges from 30%-100% in naïve animals. Was reconstitution determined prior to challenge, and if so, how did the authors deal with the variation with regard to grouping animals for the infection experiment?

Regarding controls, only naïve mice were included. From the M&M it seems naïve mice were not inoculated at all, a proper control would have been injection with PBS, give that injection of a relatively large volume of 50uL will have an effect on the structure of the tissue, and possibly the host response. In addition, particularly for the mesenchymal cell analyses, it would have been interesting to include data from lung-only animals. I believe the authors have this data available from their previous publication (Cell Rep. 2022 Apr 19;39(3):110714.).

From the text it is unclear what number of animals, donors and lungs tissues the results are based on. In M&M it says 3 donors were used, but not clear if these were used for each experiment, or whether in order to perform all experiments, a total of 3 donors were used. In the figure legends, it says the data are representative of 2 or 3 independent experiments. Is the data then represented as an average of the 2-3 experiments, or is data from 1 experiment used as representative for the 2-3 experiments? In the legend it also states that n=6-10 per time point, is this the number of animals, or the number of tissues? Is that than per experiment, or combined from 2-3 experiments. This is important because this could mean that data could be from 1 animal with 2 lung tissues per donor from 3 donors =6, or 3 animals from 1 donor.

Overall, the introduction is very long and could be reduced by summarizing from line 117-146, as this reads more as a discussion than introduction.

Reviewer #2: See above. No major experiments are required.

Reviewer #3: Kenney, et al. utilize a model of combined immune and fetal lung xenograft in mice to investigate the human tissue and immune responses to acute SARS-CoV-2 infection. While the model is interesting and may provide insights into factors that regulate acute SARS-CoV-2 infection, it relies largely on analysis of scRNA-Seq data. Strong conclusions are drawn from this scRNA-Seq data without functional validation, including conclusions about the role of endothelial cells, fibroblasts, mDNT cells, and CD4+ monocytes with little validation. Signficantly more work should be done in one or more of these areas to validate claims made. Alternatively, specific validation of one of these conclusions with appropriate tempering of the conclusion of the role of these immune factors in regulating SARS-CoV-2 clearance throughout the manuscript would strengthen this report.

The authors claim that there is an overall reduction in T cells/ILCs, but infiltration of a “viral RNA-enriched T cell population displaying myeloid-like features in infected lung tissues.” If they’re not being grouped in with ILCs/T cells, are these MAIT cells or NK T cells? Some amount of validation should be done to determine what these cells are. Is it possible this signature is a result of a technical issue with cell collection (doublets) or analysis. Is it possible myeloid cells are taking up other infected cells leading to this signature? Are these cells productively infected?

Further, are T cells actually leaving the tissue as the authors suggest or is the change in frequency as shown by scRNA-Seq due to infiltration of other cell subsets? In this model where virus is rapidly cleared from tissue explants after day 2 p.i., are T cells required? Is appropriate T cell priming occurring in lymph nodes and is the explanted tissue sufficiently vascularized for infiltration of these cells?

The authors claim a critical role for patrolling monocytes (PIM) in regulating acute infection. These cells should be looked at to determine whether they are actually patrolling in the vasculature and provide validation of these subsets within the tissue beyond scRNA-Seq.

The conclusion but this is based on assessment of scRNA-Seq and broad depletion of CD4 compared to CD3. Based on depletion studies, the authors claim T cells are not required for protection. However, it seems T cells play a role based on 50% of anti-CD3 treated tissues still containing virus on day 12 p.i. compared to 73% with anti-CD4 treatment. More specific assessment of these CD4+ monocyte populations and more specific depletion strategies should be used to justify this claim given the majority of the “persistence” of virus at day 12 p.i. is attributed to anti-CD3 treatment.

Does treatment with OKT3 or OKT4 lead to clearance in all infected explant lungs? Additionally, only day 12 is assessed. Are there earlier changes in virus titers or delays in clearance in all antibody depletion treatments?

In the last figure and at the end of the results and in the discussion, the authors show and discuss “ExiMO” but this subset is not defined in any manner. This should be clarified. In general whether monocytes or specific subsets of monocytes are important in mediating protection should be addressed more specifically through scientific data and discussion of the existing literature.

**Part III – Minor Issues: Editorial and Data Presentation Modifications**

Reviewer #1: Line 159. Please provide a rationale for using the 2019-nCoV/USA_WA1/2020 strain, as this has not been circulating for several years.

Line 196. The authors should also test for non-neutralizing antibodies against S or N, for example, which are typically present in higher concentrations and may still play a role in clearance.

Line 204. While interesting, a link with adaptation seem an overreach. The authors should also sequence virus from murine tissue if indeed this is a matter of adaptation.

Line 250. These data suggest the potential infection of T cells. Is there a way to confirm whether active replication is taking place in these cells? Could you test for sgRNA in PBMC for example?

Line 393. Is there evidence of viral RNA in endothelial cells?

Line 688. Please specify at what week post implantation, the reconstitution was assessed.

Reviewer #2: See above. No major experiments are required.

Reviewer #3: Critical details regarding the model are not included. The supplemental methods section contains the statement: “Human immune reconstitution was determined by flow cytometry at weeks post implantation.” However, the timepoint at which reconstitution by assessment of blood is not determined. Additionally, how many weeks post implantation infection is occurring at is not included in the methods.

In several instances conclusions are made from single histological images without any type of quantification 9. These include: quantification of SARS-CoV-2 IHC staining across multiple tissues in Figure 1 E-L, Figure 2 C-D describing features of damage with no broader naïve controls, quantification of SARS-CoV-2 IHC staining across multiple tissues in Figure 9C (in this case OKT3 image shows no viral antigen with Figure 9C shows half the animals still have viral antigen), and quantification of MHC-I IHC staining across multiple tissues in Figure 10.

PLOS authors have the option to publish the peer review history of their article (what does this mean? ). If published, this will include your full peer review and any attached files.

**Do you want your identity to be public for this peer review?** For information about this choice, including consent withdrawal, please see our Privacy Policy .

Reviewer #1: No

Reviewer #2: No

Reviewer #3: No

**Figure resubmission:**
---

## [Decision Letter · Decision Letter 1]

18 Aug 2025

Dear Assistant Professor Douam,

We are pleased to inform you that your manuscript 'Immune Signatures of SARS-CoV-2 Infection Resolution in Human Lung Tissues' has been provisionally accepted for publication in PLOS Pathogens.

Best regards,

Thomas E. Morrison

Academic Editor

PLOS Pathogens

Alexander Gorbalenya

Section Editor

PLOS Pathogens

Sumita Bhaduri-McIntosh

Editor-in-Chief

PLOS Pathogens

orcid.org/0000-0003-2946-9497

Michael Malim

Editor-in-Chief

PLOS Pathogens

orcid.org/0000-0002-7699-2064

Reviewer Comments (if any, and for reference):

Reviewer's Responses to Questions

**Part I - Summary**

Reviewer #3: The authors have thoroughly addresses reviewer comments to provide additional methodological details, clarify several points related to findings, and adjust language to better reflect the potential implications of results for disease based on scRNA-seq without functional validation. Given the complexities of model, it is understandable that no additional experiments were completed.

**Part II – Major Issues: Key Experiments Required for Acceptance**

Reviewer #3: None

**Part III – Minor Issues: Editorial and Data Presentation Modifications**

Reviewer #3: None

PLOS authors have the option to publish the peer review history of their article (what does this mean? ). If published, this will include your full peer review and any attached files.

**Do you want your identity to be public for this peer review?** For information about this choice, including consent withdrawal, please see our Privacy Policy .

Reviewer #3: No

---

## [Editor Report · Acceptance letter]

Dear Assistant Professor Douam,

We are delighted to inform you that your manuscript, " 

Immune Signatures of SARS-CoV-2 Infection Resolution in Human Lung Tissues," has been formally accepted for publication in PLOS Pathogens.

Best regards,

Sumita Bhaduri-McIntosh

Editor-in-Chief

PLOS Pathogens

orcid.org/0000-0003-2946-9497

Michael Malim

Editor-in-Chief

PLOS Pathogens

orcid.org/0000-0002-7699-2064